# Do Text Edits Generalize to Visual Generation?
# Benchmarking Cross-Modal Knowledge Editing in UMMs

**Xin Gao** [* 1]   **Cheng Yang** [* 1]   **Chufan Shi** [* 2]   **Taylor Berg-Kirkpatrick** [1]

## Abstract

Unified multimodal models (UMMs) have emerged as a promising paradigm for general-purpose multimodal intelligence. As they are deployed in real-world applications, effectively updating internal knowledge becomes critical. While knowledge editing has matured for text-only models, it remains unclear whether edits that successfully modify textual outputs also transfer to image generation in UMMs. To study this question, we introduce UNIKE, the first benchmark for cross-modality knowledge editing in UMMs, comprising 2,971 edit subjects spanning attribute and relation edits. Using VQA-based visual verification, we reveal a striking modality gap: text-side efficacy can reach approximately 92%, whereas the best overall VQA accuracy under direct image generation is only 18.5%. We further propose Reasoning-augmented Parameter Editing, which explicitly activates edited knowledge before generation and improves overall VQA accuracy for all evaluated model-editor pairs, with gains up to 18.6 percentage points. Mechanistic analysis shows that this gap is associated with partial alignment between edited textual representations and the conditioning pathways for visual generation, where edits sufficient for text outputs may remain too weak or misaligned to steer image synthesis. These findings show that textual knowledge edits do not guarantee reliable cross-modality transfer and motivate modality-aware editing methods. Our code and data are available at https://github.com/gxx27/UniKE.

---
[*]Equal contribution  [1]University of California San Diego  [2]University of Southern California. Correspondence to: Xin Gao <xig022@ucsd.edu>, Cheng Yang <chy085@ucsd.edu>, Chufan Shi <chufansh@usc.edu>, Taylor Berg-Kirkpatrick <tberg@ucsd.edu>.

*Proceedings of the 43rd International Conference on Machine Learning*, Seoul, South Korea. PMLR 306, 2026. Copyright 2026 by the author(s).

## 1. Introduction

Unified multimodal models (UMMs) have emerged as a promising paradigm for general-purpose multimodal intelligence (Deng et al., 2025; Wang et al., 2025; Yang et al., 2026; Zhou et al., 2024). Unlike conventional pipelines that rely on cascaded components (Lian et al., 2024; Shen et al., 2023; Wu et al., 2023), UMMs adopt a unified backbone that represents images and text jointly, enabling seamless parameter sharing across understanding and generation (Zhang et al., 2025; 2026). By inheriting world knowledge and reasoning capabilities from textual pretraining, UMMs can leverage these priors to guide visual content synthesis, moving us closer to foundation models that both interpret and simulate the physical world through a unified interface.

As UMMs are increasingly deployed in real-world applications, the ability to effectively update their internal knowledge becomes a critical concern. In the textual domain, knowledge editing (KE) (Wang et al., 2023a; Zhang et al., 2024b), which aims to modify specific facts without retraining from scratch, has emerged as a solution to this challenge. However, despite the maturity of techniques for language models, knowledge editing for UMMs remains largely unexplored. Given that UMMs unify text and image generation within a shared backbone, a natural question arises: ***if we edit knowledge solely through the textual modality, will such edits propagate to visual generation, enabling the model to synthesize images that reflect the updated knowledge?*** Concretely, as illustrated in Figure 1(a), if we edit the model to assert "an apple is blue" in text, will it generate a blue apple when prompted for image generation?

To systematically investigate this question, we establish UNIKE, the first benchmark tailored for cross-modality knowledge editing in UMMs. In UNIKE, we apply text-side knowledge editing and test whether the updated knowledge is expressed consistently in visual generation. The benchmark contains 2,971 carefully constructed edit subjects, covering two axes of factual knowledge: attribute and relation. These edit subjects yield 5,535 evaluation instances after expanding them into stage-specific and visually verifiable generation settings. Since fine-grained factual correctness in generated images is difficult to evaluate with generic text-to-image metrics, we use VQA-based visual verification to

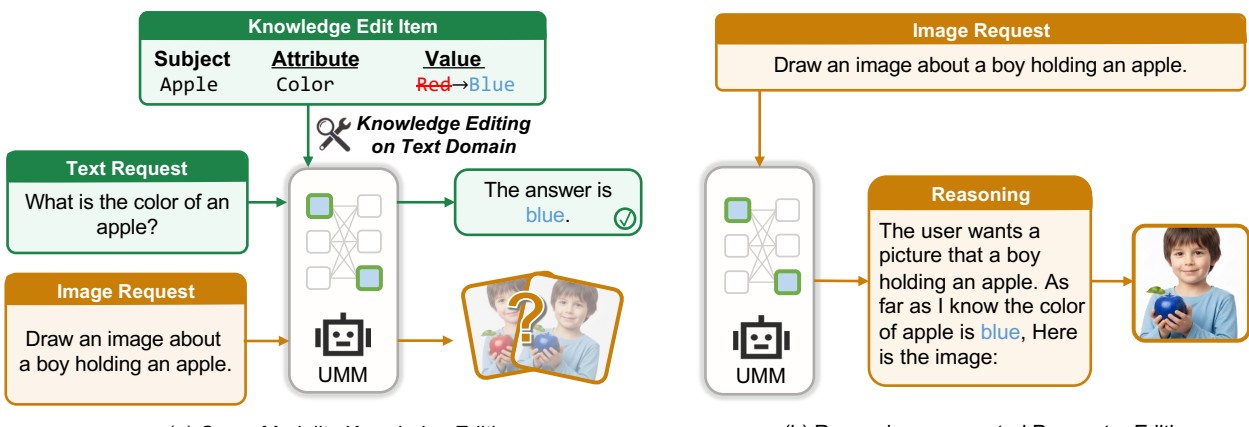

(a) Cross-Modality Knowledge Editing  (b) Reasoning-augmented Parameter Editing

*Figure 1.* **Cross-modality knowledge editing in unified multimodal models.** We edit a UMM to change an attribute (i.e., *apple color: red → blue*). (a) Cross-modality knowledge-editing under-explored: While text-domain editing successfully updates the model's textual answers, the propagation of this updated knowledge to visual generation remains under-explored. (b) Reasoning-augmented Parameter Editing: By eliciting an explicit reasoning step, the model activates the latent edited knowledge and transfers it to image generation.

assess whether each generated image is consistent with the post-edit knowledge.

Empirical results on UNIKE reveal a striking cross-modality knowledge-editing gap: parameter-editing methods that substantially improve textual outputs yield much weaker and less reliable changes in image generation. To mitigate this gap, we introduce Reasoning-augmented Parameter Editing (Figure 1(b)). Our key hypothesis is that edited knowledge stored in parameters may remain latent for image generation unless it is explicitly activated in the textual context. By eliciting an intermediate textual reasoning step prior to image generation, we encourage the model to verbalize the edited fact and therefore provide a stronger semantic constraint for subsequent visual synthesis.

We further provide mechanistic evidence for why high text-side edit success does not reliably translate into visual changes. Our analyses point to a conditioning-pathway bottleneck: edit-induced perturbations that are sufficient to alter textual recall are not necessarily preserved or expressed in the representations that condition image generation. In architectures with an explicit projection interface, this bottleneck can attenuate edit-induced signals before they reach the image generator; in architectures without such a projection, the delivered perturbations can still remain insufficient to reliably override strong visual priors. The reasoning-augmented protocol partially mitigates this issue by introducing an explicit textual conditioning shift that is larger and qualitatively different from the direct edit, thereby improving edited knowledge transfer across modalities.

To sum up, we highlight our contributions as follows:

- We introduce UNIKE, the first benchmark for cross-

modality knowledge editing in UMMs, covering both attribute- and relation-centric edits. The benchmark contains 2,971 edit subjects and 5,535 evaluation instances designed to test whether text-side edits are reflected in visually verifiable image generations.

- We systematically evaluate parameter-editing baselines and demonstrate a substantial modality gap: edits that are successful under text-only metrics do not necessarily induce consistent visual changes. We further propose Reasoning-augmented Parameter Editing, which explicitly activates edited knowledge through an intermediate reasoning step before image generation.

- We provide mechanistic analyses showing that limited cross-modality transfer is associated with a conditioning-pathway bottleneck. Edit-induced perturbations can be attenuated or weakly expressed before reaching the image generator, while reasoning augmentation introduces a stronger textual conditioning shift that partially improves visual grounding.

## 2. Related Work

**Knowledge Editing.** Updating the factual beliefs of large models without expensive retraining is a critical challenge. Current approaches primarily branch into parameter-preserving methods, such as MemPrompt (Madaan et al., 2022), MeLLo (Zhong et al., 2023), IKE (Zheng et al., 2023a), which utilizes in-context demonstrations to guide model behavior, and parameter-modifying methods like ROME (Meng et al., 2022), MEMIT (Meng et al., 2023), PMET (Li et al., 2024), and AlphaEdit (Fang et al., 2025a), which locate and alter specific neural weights associated with factual recall. Extending this to the visual domain, Text-to-Image (T2I) editing methods such as TIME (Orgad

*Table 1.* Comparison with representative knowledge editing benchmarks. T2T denotes text-to-text evaluation, T2I denotes text-to-image generation, and I2T denotes image-conditioned text answering. Size reports the number of edit cases.

| Benchmark | Primary Modality | Task | Evaluation Path | Visually Verifiable | Size |
|---|---|---|---|---|---|
| ZsRE (Levy et al., 2017) | Text | Factual QA | T2T | ✗ | 10k |
| CounterFact (Meng et al., 2022) | Text | Factual rewriting | T2T | ✗ | 21k |
| MQuAKE (Zhong et al., 2023) | Text | Multi-hop QA | T2T | ✗ | 3.0k |
| TMKE (Fang et al., 2025b) | Image+Text | Cross-modal QA | I2T | ✗ | 3.7k |
| **UNIKE** | **Text+Image** | **Cross-modal generation** | **T2T+T2I** | ✓ | **3.0k** |

et al., 2023), ReFACT (Arad et al., 2024), and DiffQuick-Fix (Basu et al., 2024) have been proposed to correct visual concepts. However, these T2I methods predominantly operate on modular architectures (Rombach et al., 2021) by targeting distinct text encoders or cross-attention layers, rather than the monolithic backbones found in unified systems.

**Knowledge Editing Benchmarks.** A number of benchmarks have been proposed to evaluate knowledge editing in language and multimodal models. ZsRE (Levy et al., 2017) evaluates text-based question-answering edits and robustness to rephrased queries. CounterFact (Meng et al., 2022) focuses on counterfactual subject–relation–object rewrites with locality probes, while MQuAKE (Zhong et al., 2023) tests whether edited facts propagate to multi-hop textual reasoning. More recently, TMKE (Fang et al., 2025b) studies cross-modal consistency in multimodal knowledge editing, primarily through image-conditioned text answering. In contrast, UNIKE evaluates a novel pathway: after applying text-side edits to a UMM, it tests whether the edited knowledge appears in generated images.

**Unified Multimodal Models.** Unlike modular frameworks that decouple understanding and generation, UMMs (Cui et al., 2025; Deng et al., 2025; Wang et al., 2025) integrate both textual and visual modalities within a single framework (Liang et al., 2025). These architectures typically employ a shared transformer backbone that aligns visual representations via quantization or diffusion heads with text tokens, often utilizing a unified autoregressive objective or flow-matching mechanism (Liang et al., 2025). However, the mechanisms of knowledge storage and cross-modal interaction within UMMs remain opaque. Consequently, it is unclear how textual knowledge editing impacts the image generation process, specifically whether factual updates can effectively transcend modality boundaries (Tao et al., 2026).

## 3. The UNIKE Benchmark

### 3.1. Task Definition

In this work, we study cross-modality knowledge editing in unified multimodal models. After applying a text-side

edit to change a model's textual response for a fact, we test whether the updated knowledge is expressed consistently in visual generation and can be verified automatically.

Each evaluation instance in UNIKE specifies a statement-completion edit target $q$ with a pre-edit completion $y$ and a target completion $y'$. It also provides an image generation prompt $p_{\text{img}}$, a visual target description $t_{\text{vis}}$ describing the observable implication of $y'$, and a VQA query $q_{\text{vqa}}$ for verification. We represent an evaluation instance as:

$$\mathcal{I} = \big(q,\ y,\ y',\ p_{\text{img}},\ t_{\text{vis}},\ q_{\text{vqa}}\big). \tag{1}$$

Figure 2 presents some concrete UNIKE instances.

The benchmark covers two families of edits. Attribute edits modify intrinsic visual properties of an entity, like color or pattern, and are instantiated with progressively more compositional image prompts. Relation edits modify relational facts about an entity and are restricted to categories with depictable implications in images, such as location or occupation, so that the edited relation can be verified through characteristic visual evidence. In all cases, success requires that the generated image satisfies $t_{\text{vis}}$ and that answering $q_{\text{vqa}}$ over the image is consistent with the post-edit completion $y'$. Table 1 summarizes the key differences between UNIKE and representative knowledge editing benchmarks.

### 3.2. Benchmark Construction

We construct UNIKE following a systematic pipeline organized around a single key constraint: *visualizability*. This constraint ensures that each target completion $y'$ implies a concrete visual outcome that can be reliably verified through VQA evaluation. Image prompts are designed to be answer-neutral: they depict the subject or scene without revealing either the original or edited value, so that any correct visual output must originate from the model's edited internal knowledge rather than prompt-based shortcuts. Figure 3 illustrates the resulting distributions across both edit families.

### 3.3. Attribute Edits

We generate attribute-edit candidates via a self-instruction style pipeline (Wang et al., 2023b) using Gemini-3.0-Flash (Gemini Team, Google, 2025). Each candidate specifies an object and an attribute domain drawn from color,

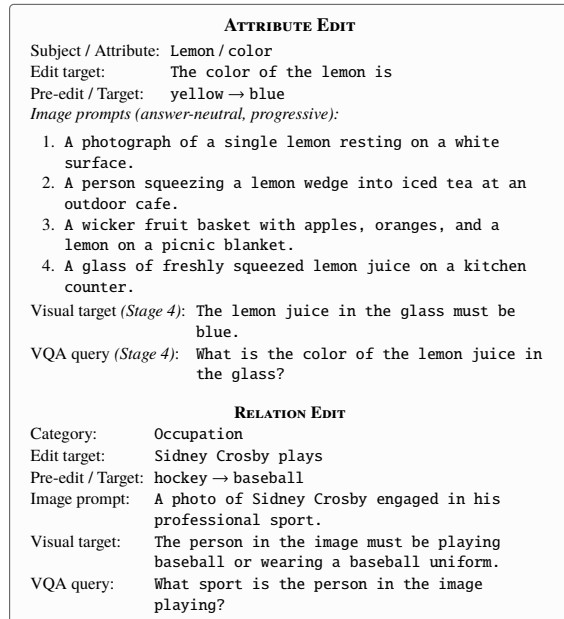

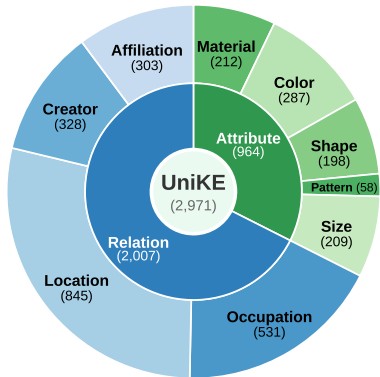

*Figure 3.* Benchmark composition. Editing tasks are split into two categories (inner ring): attribute (material, color, shape, pattern, and size), and relation (affiliation, creator, location, and occupation). The outer ring shows the subcategory distribution.

*Table 2.* Attribute edit stages and counts. Counts denote stage-specific evaluation instances after visual-testability filtering. Stages increase compositional difficulty while keeping the image prompt answer-neutral.

| Stage | Num. | Description |
|---|---|---|
| 1 | 959 | Direct attribute reference. |
| 2 | 874 | Entity in a realistic context or scene. |
| 3 | 858 | Multi-entity scene with interactions. |
| 4 | 837 | Derived product/use inherits the attribute. |

*Figure 2.* Example data instances from UNIKE, illustrating the structure of attribute edits across stages and relation edits.

material, shape, size, and pattern, together with a pre-edit completion $y$ and a counterfactual target completion $y'$. The generation prompt enforces answer-neutral image prompts that depict the subject without revealing either the original or edited attribute value and non-tautological visual targets with category-specific constraints. For the pattern domain, we employ a curated subject pool of entities with characteristic surface textures, paired with balanced targets drawn from a lexicon of common pattern adjectives, to avoid degenerate distributions in target selection.

**Stage Design.** To assess cross-modality knowledge editing for UMMs under increasing difficulty, we instantiate each edit into up to four stages with progressively stricter demands on grounding and compositional generalization:

1. Atomic object. Queries the edited attribute under a direct reference prompt, where the image prompt depicts the entity without revealing the edited value.
2. Realistic context. Embeds the entity in a realistic context and tests whether the edited attribute remains stable under variations in lighting, viewpoint, and background.
3. Complex composition. Introduces complex multi-entity scenes with interactions, where the model must localize the correct entity and apply the edited attribute despite competing visual cues.
4. Derived product. Shifts the target to derivatives or direct uses that should inherit the edited attribute, requiring attribute transfer to a related visual form. This represents the most challenging setting for text-side editing because the prompt queries the attribute through a derived refer-

ent rather than the edited entity itself.

**Filtering Criteria.** To ensure visual verifiability, each generated stage is independently assessed for visual testability: stages whose subject–target pair cannot produce a visually distinguishable outcome are dropped. All retained instances undergo automated validation that enforces answer-neutrality in image prompts, non-tautological visual targets, correct subject-naming conventions in VQA queries, and category-specific constraints. This process yields 964 attribute edit items across all domains, corresponding to 3,528 stage-specific attribute evaluation instances. Table 2 reports the volume per stage, and Figure 3 summarizes the attribute-domain distribution. Prompt templates and generation details are provided in Appendix F.1.

### 3.4. Relation Edits

For relation edits, we draw statement-completion triples $(q, y, y')$ from widely used knowledge-editing benchmarks, including CounterFact (Meng et al., 2022) and MQuAKE (Zhong et al., 2023). We convert each triple into our cross-modality schema by generating an answer-neutral image prompt $p_{\mathrm{img}}$, a visual target description $t_{\mathrm{vis}}$, and a VQA query $q_{\mathrm{vqa}}$. The image prompt is designed to elicit a depiction relevant to the edited relation without directly revealing either the original or target completion.

**Filtering Criteria.** We filter relation edits through a two-stage pipeline. First, we apply rule-based filtering to remove candidates with abstract or non-visualizable relations, such as citizenship, etymology, and native language; entries where the original answer is lexically embedded in the subject name; generic position-as-subject entries; and exact duplicates. Second, we use an LLM-as-a-judge framework (Zheng et al., 2023b) with Gemini-3.0-Flash (Gemini Team, Google, 2025) to retain only relations whose edited target can be grounded in a depictable visual category, including affiliation, creator, location, and occupation. For retained items, we validate that the generated image prompts remain answer-neutral, that the visual targets are concretely verifiable, and that the VQA queries refer to the intended subject unambiguously.

### 3.5. Benchmark Summary

The final UNIKE benchmark contains 2,971 edit subjects: 964 attribute edits with progressive stages (Table 2) and 2,007 relation edits, yielding 5,535 total evaluation instances with multi-stage design, after visualizability filtering and automated validation.

## 4. Evaluation of Cross-Modality Knowledge Editing

Standard knowledge-editing evaluations primarily measure whether an edited model produces the target answer in text. For UMMs, text-side success alone is insufficient: the edited knowledge must also be expressed in generated images when the image prompt does not reveal the target answer. To quantify this cross-modality transfer, we evaluate edited models under two protocols: a direct generation protocol that tests whether the edit implicitly affects visual synthesis, and a reasoning-augmented protocol that first elicits the edited knowledge in text before image generation.

**DIRECT.** The model generates an image from the image prompt $p_{\text{img}}$ alone. This protocol tests whether the parameter edit itself is sufficient to affect image generation.

**REASONING-AUGMENTED.** Before image generation, the edited model is prompted with a category-conditioned reasoning template $p_{\text{rea}}$ to produce an intermediate textual rationale $r$. The rationale is then provided as an additional textual constraint alongside the same image prompt $p_{\text{img}}$ using a fixed formatting rule. The reasoning templates are defined at the category level and applied uniformly to all instances in the same category. They are designed to recall the relevant edited knowledge, connect it to the subject in the image prompt, and express the result as a visual description. Importantly, the rationale is generated by the edited model itself rather than supplied as oracle ground truth; therefore,

this protocol measures whether explicit activation of edited knowledge improves visual transfer.

Both protocols use the same image prompt $p_{\text{img}}$ and the same VQA-based verification procedure. They differ only in whether the model-generated rationale $r$ is included as additional conditioning during image generation.

### 4.1. Evaluation Metrics

We report three metrics that correspond to key components of the evaluation protocols: text-side edit efficacy, edited-knowledge reasoning accuracy under the REASONING-AUGMENTED protocol, and visual correctness verified by VQA. We report the average score over the evaluation set.

**Text-side Efficacy.** Text-side efficacy measures whether the edited model produces the target completion $y'$ for the edit prompt $q$ in a text-only setting. Following prior work (Fang et al., 2025a), we adopt a lightweight next-token criterion. Let $\tau(\cdot)$ denote tokenization, and let $\hat{t}(q)$ be the model's most likely next token given $q$. We compute

$$\text{Eff.} = \frac{1}{N} \sum_{i=1}^{N} \mathbb{I}\left[\hat{t}(q_i) = \tau(y'_i)[1]\right]. \quad (2)$$

**Reasoning Accuracy.** Reasoning Accuracy measures whether the intermediate rationale $r$ produced under the REASONING-AUGMENTED protocol explicitly states the edited completion $y'$. Let $\text{norm}(\cdot)$ denote a simple text normalization, and let $\preceq$ denote substring containment. We compute the following score

$$\text{Acc}_{\text{Reason}} = \frac{1}{N} \sum_{i=1}^{N} \mathbb{I}[\text{norm}(y'_i) \preceq \text{norm}(r_i)]. \quad (3)$$

**VQA Accuracy.** VQA Accuracy measures whether a generated image $x$ expresses the edited visual implication specified by $t_{\text{vis}}$ and can be verified by answering the associated query $q_{\text{vqa}}$. We use Qwen3-VL-235B-A22B-Instruct (Bai et al., 2025) as an automated judge that outputs a binary decision $J(x, q_{\text{vqa}}, t_{\text{vis}}) \in \{0, 1\}$. We calculate

$$\text{Acc}_{\text{VQA}} = \frac{1}{N} \sum_{i=1}^{N} J(x_i, q_{\text{vqa},i}, t_{\text{vis},i}). \quad (4)$$

The judge's prompt is provided in Appendix F.3.

### 4.2. Experimental Setup

We evaluate cross-modality knowledge editing on three representative unified multimodal models with distinct architectures: Ovis-U1 (Wang et al., 2025), BLIP3o-4B (Chen et al., 2025), and OmniGen2 (Wu et al., 2026). For each model,

*Table 3.* Main results on cross-modality knowledge editing. **Eff.** denotes text-side efficacy; **Reason.** denotes reasoning accuracy under the REASONING-AUGMENTED protocol; **VQA** denotes VQA accuracy. Overall is the average over all evaluated edit subjects. In DIRECT, the best result in each column per model is underlined; in REASONING-AUGMENTED, the best result in each column per model is bolded. *For models with shared visual backbones, AlphaEdit uses a softened null-space projector; see Appendix C for details.

| Model | Method | Attribute | | | Relation | | | Overall | | |
|---|---|---|---|---|---|---|---|---|---|---|
| | | Eff. | Reason. | VQA | Eff. | Reason. | VQA | Eff. | Reason. | VQA |
| DIRECT (NO REASONING) | | | | | | | | | | |
| Ovis-U1 | MEMIT | 49.01 | – | 7.40 | 65.02 | – | 9.32 | 59.84 | – | 8.70 |
| | PMET | 55.16 | – | 8.65 | 80.32 | – | 10.21 | 72.18 | – | 9.71 |
| | AlphaEdit | 38.69 | – | 7.09 | 77.18 | – | 7.57 | 64.73 | – | 7.42 |
| BLIP3o-4B | MEMIT | 51.82 | – | 12.30 | 77.03 | – | 16.54 | 68.88 | – | 15.17 |
| | PMET | 50.89 | – | 13.35 | 88.44 | – | 20.98 | 76.30 | – | 18.51 |
| | AlphaEdit* | 51.62 | – | 13.24 | 90.43 | – | 17.49 | 77.88 | – | 16.12 |
| OmniGen2 | MEMIT | 34.41 | – | 5.74 | 69.71 | – | 10.16 | 58.29 | – | 8.73 |
| | PMET | 48.28 | – | 7.92 | 89.54 | – | 13.10 | 76.20 | – | 11.43 |
| | AlphaEdit* | 43.80 | – | 8.03 | 91.93 | – | 13.15 | 76.37 | – | 11.50 |
| REASONING-AUGMENTED | | | | | | | | | | |
| Ovis-U1 | MEMIT | 49.01 | 42.96 | 24.61 | 65.02 | 43.90 | 24.31 | 59.84 | 43.59 | 24.41 |
| | PMET | **55.16** | **44.53** | **27.42** | **80.32** | **57.90** | **28.75** | **72.18** | **53.57** | **28.32** |
| | AlphaEdit | 38.69 | 35.25 | 20.54 | 77.18 | 48.88 | 20.73 | 64.73 | 44.47 | 20.67 |
| BLIP3o-4B | MEMIT | **51.82** | 27.95 | 15.85 | 77.03 | 53.91 | 16.54 | 68.88 | 45.52 | 16.32 |
| | PMET | 50.89 | 34.31 | 16.16 | 88.44 | 62.23 | **20.78** | 76.30 | 53.20 | **19.29** |
| | AlphaEdit* | 51.62 | **39.42** | **16.37** | **90.43** | **72.30** | 17.79 | **77.88** | **61.67** | 17.33 |
| OmniGen2 | MEMIT | 34.41 | 30.14 | 7.40 | 69.71 | 51.57 | 14.45 | 58.29 | 44.64 | 12.17 |
| | PMET | **48.28** | 31.91 | 8.24 | 89.54 | 61.53 | **19.73** | 76.20 | 51.96 | 16.01 |
| | AlphaEdit* | 43.80 | **36.91** | **14.39** | **91.93** | **69.46** | 19.58 | **76.37** | **58.93** | **17.90** |

we apply three representative parameter-editing methods, MEMIT (Meng et al., 2023), PMET (Li et al., 2024), and AlphaEdit (Fang et al., 2025a), under both the DIRECT and REASONING-AUGMENTED protocols in a sequential editing setting. Following prior findings that factual knowledge in generative language models is primarily localized in middle MLP layers (Meng et al., 2022), we restrict parametric updates to the intermediate layers of the text-side backbones. Specifically, we edit layers 4–8 for Ovis-U1 and layers 6–10 for BLIP3o-4B and OmniGen2.

For BLIP3o-4B and OmniGen2, the Qwen2.5-VL backbone is shared across text understanding and visual conditioning. In this setting, the standard AlphaEdit null-space projection can overly constrain parameter updates and degrade generative behavior. We therefore use a softened null-space projector by interpolating the AlphaEdit projector with the identity matrix, with interpolation coefficient $\alpha$ set to 0.7 for BLIP3o-4B and 0.6 for OmniGen2. We mark these modified AlphaEdit runs with an asterisk in Table 3. Further implementation details are provided in Appendix C.

### 4.3. Results on Cross-Modality Knowledge Editing

Table 3 reports the main results under the DIRECT and REASONING-AUGMENTED protocols. The most salient pattern is a large modality gap. Under direct generation,

models often achieve strong text-side efficacy, but only a small fraction of these edits are reflected in visually verifiable images. Across the best direct settings for each model, VQA accuracy retains only about one eighth to one quarter of the corresponding text-side efficacy. This shows that changing the model's textual completion behavior is far from sufficient to steer image synthesis (Luo et al., 2025; 2026; Shi et al., 2026).

Reasoning-augmented generation narrows this gap, but the gains are architecture- and category-dependent. Overall VQA accuracy improves for every model-editor pair, with the largest relative gains on Ovis-U1 and more modest gains on BLIP3o-4B and OmniGen2. This suggests that reasoning is most useful when edited knowledge is present in the model but is not naturally exposed to the visual-conditioning pathway by a neutral image prompt. By first verbalizing the edited fact, the model converts a latent parameter change into an explicit textual constraint for generation, consistent with observations that reasoning interventions can reshape multimodal representations and improve downstream visual behavior (Luo et al., 2025; 2026; Shi et al., 2026). Nevertheless, reasoning does not eliminate the gap, indicating that textual recovery and visual realization remain distinct.

Across methods, performance follows a consistent cascade from text-side efficacy to reasoning accuracy and then to

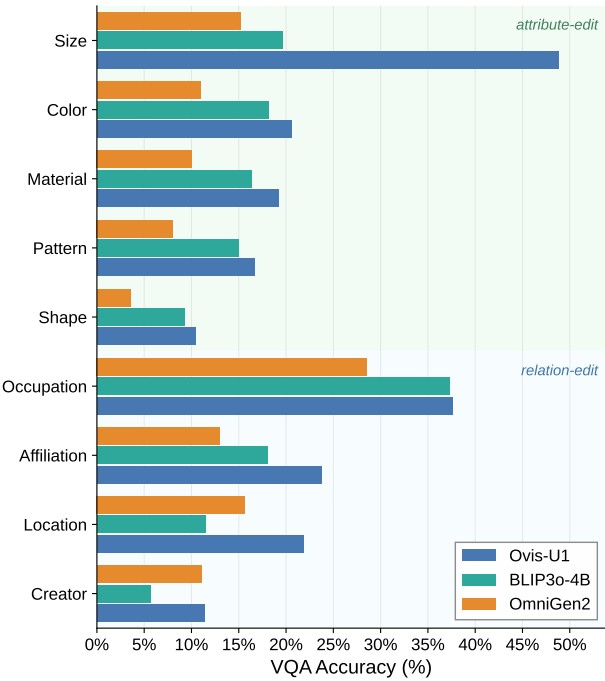

*Figure 4.* VQA accuracy under the REASONING-AUGMENTED protocol, broken down by attribute domains (light green background) and relation categories (light blue background).

**VQA accuracy.** The first drop reflects limited generalization of parameter edits beyond the canonical edit prompt, in line with prior findings that edits can be overly localized and brittle under paraphrases or neighboring queries (Fang et al., 2025a). The second drop reflects the cross-modal bottleneck: even when the edited target is recovered in text, image generation must still bind it to the correct visual referent, overcome pretrained visual priors, and render evidence clearly enough for VQA verification.

Attribute and relation edits fail in different ways. Relation edits generally achieve higher text-side efficacy because they often resemble discrete subject–object substitutions. Attribute edits are harder to edit textually, but once retrieved, their visual implications are often direct, such as color, material, or size. In contrast, relation edits may require distinctive contextual evidence, landmarks, artifacts, logos, or identity cues, making them difficult to verify visually even when the edited relation is recalled. These results show that cross-modal knowledge editing requires not only successful textual recall, but also reliable transmission of the edited knowledge into visually grounded generation.

### 4.4. Stage-wise Analysis

Figure 5 reports stage-wise results for attribute edits under the REASONING-AUGMENTED protocol. The clearest pattern is that text-side efficacy drops sharply as soon as the prompt departs from the canonical edit form. Averaged

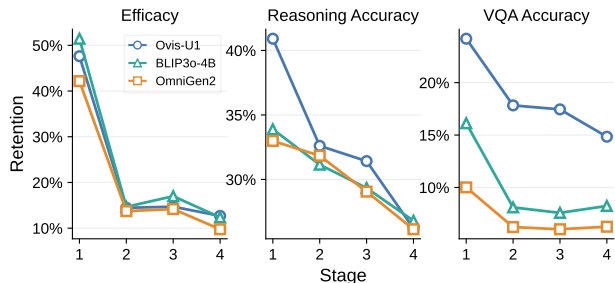

*Figure 5.* Attribute-stage retention under the REASONING-AUGMENTED protocol. Text-side efficacy, reasoning accuracy, and VQA accuracy are reported across Stages 1–4 and normalized by the Stage 1 subset size.

across models, moving from Stage 1 to Stage 2 reduces efficacy by roughly 70%, even though the edited entity and attribute remain explicit. The additional change from Stage 2 to Stage 3 is comparatively small, while Stage 4 causes another decline. This suggests that the primary text-side failure mode is not scene complexity itself, but sensitivity to the original edit trigger. Derived-product prompts are especially difficult because they require transferring the edited attribute through an indirect referent rather than recalling it from the edited entity.

Reasoning accuracy is substantially more stable across stages. Across models, it loses only about one tenth of its Stage 1 value at Stage 2 and about one quarter by Stage 4. This indicates that the reasoning prompt acts as a more robust retrieval interface: it partially recovers the edited attribute even when the stage-specific prompt no longer matches the original editing template. The contrast between the steep efficacy drop and the milder reasoning drop suggests that many edits are not completely absent from the model, but are difficult to elicit under contextual or compositional rephrasings.

However, this recovered textual knowledge is only partially converted into visual evidence. VQA accuracy decreases by roughly 35–40% from Stage 1 to the later stages on average, and remains far below reasoning accuracy throughout. The gap is particularly informative when viewed as a conversion ratio from reasoning to visual verification. Ovis-U1 preserves a relatively stable fraction of its reasoning success as VQA success across stages, whereas BLIP3o-4B and OmniGen2 show a much lower conversion rate once the prompt enters contextual or compositional settings. Thus, even when the edited attribute is verbalized, models differ substantially in their ability to bind that attribute to the correct visual referent and render it in a verifiable way.

These trends reinforce the central finding of UNIKE: textual edit success, edited-knowledge retrieval, and visual realization are separate stages of cross-modal transfer. Reasoning improves the retrieval stage, but the remaining drop to VQA

accuracy shows that visual generation introduces an additional bottleneck, especially under contextual grounding, multi-entity binding, and derived-reference generalization.

## 4.5. Category-wise Analysis

Figure 4 reports category-wise VQA accuracy under the REASONING-AUGMENTED protocol. For attribute edits, size is the easiest category across models, especially for Ovis-U1. This is because size edits are evaluated through relative comparisons with reference objects, giving both the generator and the VQA judge a concrete visual relation to ground. Color and material are moderately transferable, as they correspond to intrinsic visual properties that can be rendered directly, but they remain vulnerable to strong pre-trained object priors and ambiguous appearance. Shape is consistently the hardest attribute category: although models can often retrieve the edited shape textually, precise geometric control is difficult to realize in generation. Pattern is also challenging, mainly because surface textures require fine-grained and spatially consistent rendering, though successful cases can be visually salient once the pattern is produced.

For relation edits, occupation is the easiest category for all models. Occupations often have localized visual proxies, such as uniforms, tools, or canonical activities, which make the edited relation easier to translate into image evidence. Location and affiliation are more difficult because they depend on contextual cues such as landmarks, architecture, logos, or team-specific symbols, which generators may omit or render ambiguously. Creator is among the hardest categories despite high text-side efficacy, since authorship is rarely visually observable without explicit text labels or recognizable identity cues. Overall, the category-wise results show that cross-modal transfer is strongest when the edited fact maps to concrete, localized visual evidence, and weakest when verification depends on abstract attribution, precise geometry, or subtle contextual cues.

## 4.6. Mechanistic Analysis: The Conditioning Pathway Bottleneck

Table 3 shows that text-side edit success is only weakly predictive of visually verifiable generation. We hypothesize that this gap arises from the conditioning pathway that maps edited language representations into the visual generator. A text edit only needs to perturb the language model enough to change the next-token distribution, whereas image generation depends on whether the resulting hidden-state change is preserved in the representations that condition the diffusion transformer (DiT). We therefore analyze the conditioning signal received by the DiT under both the DIRECT and REASONING-AUGMENTED protocols.

*Table 4.* Conditioning-pathway measurements for PMET on 100 sampled cases. The first two columns measure implicit edit-induced drift at the LLM output under the same neutral image prompt. The last two columns measure the effective DiT-input conditioning distance under DIRECT and REASONING-AUGMENTED generation.

| Model | Edit drift | | Protocol distance | |
|---|---|---|---|---|
| | $d_{\cos}^{\text{tok}}$ | $r_F$ | $d_{\cos}^{\text{dir}}$ | $d_{\cos}^{\text{rea}}$ |
| Ovis-U1 | 0.003 | 0.078 | 0.018 | 0.154 |
| BLIP3o-4B | 0.139 | 0.527 | 0.031 | 0.064 |
| OmniGen2 | 0.038 | 0.262 | 0.018 | 0.092 |

**Experimental design.** All three evaluated UMMs use an LLM to encode textual input and a DiT to generate images, but they differ in how language representations are converted into visual conditioning. The architectural differences are summarized in Appendix D.1. We use cosine offset as the basic unit for measuring representation change. For two vectors $a$ and $b$, define

$$\Delta_{\cos}(a, b) = 1 - \frac{a^\top b}{\|a\| \, \|b\|}. \tag{5}$$

Let $C_{\text{fresh}}^{\text{LLM}}$ and $C_{\text{edit}}^{\text{LLM}}$ denote the LLM-output conditioning sequences produced by the unedited and edited models for the same neutral image prompt, with $\delta = C_{\text{edit}}^{\text{LLM}} - C_{\text{fresh}}^{\text{LLM}}$. To quantify the implicit perturbation induced by editing, we compute the average per-token cosine offset

$$d_{\cos}^{\text{tok}} = \frac{1}{T} \sum_{t=1}^{T} \Delta_{\cos} \left( C_{\text{edit},t}^{\text{LLM}}, C_{\text{fresh},t}^{\text{LLM}} \right), \tag{6}$$

together with the relative Frobenius drift $r_F = \|\delta\|_F / \|C_{\text{fresh}}^{\text{LLM}}\|_F$. For Ovis-U1, this edit-drift measurement is taken before the frozen projection; for BLIP3o-4B and OmniGen2, the LLM-output representation is also the representation supplied to the DiT.

We further compare the effective conditioning shift under the two generation protocols. Since REASONING-AUGMENTED generation introduces an additional rationale and therefore changes the conditioning length, we compare mean-pooled DiT-input conditioning vectors. Let $\bar{c}_{\text{fresh}}^{\text{dir}}$ denote the mean-pooled conditioning vector of the unedited model under direct generation, and let $\bar{c}_{\text{edit}}^{\text{dir}}$ and $\bar{c}_{\text{edit}}^{\text{rea}}$ denote the corresponding vectors of the edited model under direct and reasoning-augmented generation. We compute

$$d_{\cos}^{\text{dir}} = \Delta_{\cos} \left( \bar{c}_{\text{edit}}^{\text{dir}}, \bar{c}_{\text{fresh}}^{\text{dir}} \right), \quad d_{\cos}^{\text{rea}} = \Delta_{\cos} \left( \bar{c}_{\text{edit}}^{\text{rea}}, \bar{c}_{\text{fresh}}^{\text{dir}} \right). \tag{7}$$

For Ovis-U1, these protocol-level distances are measured after the frozen projection, because this is the representation actually received by the DiT. Thus, the first measurement captures the implicit parameter-edit signal, whereas the second measurement captures how much conditioning shift

is ultimately available to the image generator under each protocol.

**Conditioning drift reveals an architectural bottleneck.**
Table 4 shows that the implicit edit signal differs markedly across architectures. Ovis-U1 has a very small LLM-output directional drift, whereas BLIP3o-4B and OmniGen2 reach 0.139 and 0.038, respectively. The same trend appears in the magnitude-based measure: Ovis-U1 reaches only 0.078, compared with 0.527 for BLIP3o-4B and 0.262 for Omni-Gen2. Thus, the edit-induced perturbation in Ovis-U1 is not merely smaller by a marginal amount; it is substantially weaker in both direction and magnitude before the visual generator is conditioned.

This pattern is consistent with Ovis-U1's frozen dimensionality-reducing projection. The SVD analysis in Appendix D.2 shows that perturbations produced by current MLP-based editors are broadly distributed rather than concentrated in the subspace preserved by this projection. As a result, the projection acts as an architectural filter: it preserves enough information for ordinary language-conditioned generation, but it can attenuate the off-distribution perturbations introduced by parameter editing (Serra et al., 2026; Tong et al., 2024). BLIP3o-4B and OmniGen2 do not contain an analogous frozen projection interface, which helps explain why their edited representations induce larger conditioning drift.

The protocol-level distances further indicate that a larger implicit drift is not sufficient by itself. BLIP3o-4B has the largest edit drift at the LLM output, but its direct DiT-input distance remains only 0.031, and its final VQA gains are limited. Conversely, Ovis-U1 has the weakest implicit edit drift, yet it benefits most from reasoning augmentation. This suggests that cross-modal transfer is governed not only by the size of the edit-induced perturbation, but also by whether the perturbation is aligned with the conditioning pathway used for image synthesis. An edit may be strong enough to change textual completion while still failing to produce a sufficiently usable conditioning shift for visual generation (Luo et al., 2026). The propagation analysis in Appendix D.3 further indicates that the dominant attenuation occurs before the DiT rather than inside the diffusion transformer itself.

**Reasoning exposes edited knowledge through a stronger conditioning pathway.** The last two columns of Table 4 show that reasoning augmentation increases the DiT-input conditioning distance for all three models, but the amplification is highly architecture-dependent. The increase is largest for Ovis-U1, where reasoning enlarges the effective conditioning shift by a factor of 8.56. OmniGen2 also shows a substantial amplification, by a factor of 5.11, whereas BLIP3o-4B exhibits a more moderate increase, by a fac-

tor of 2.06. Thus, reasoning is most beneficial for models whose direct conditioning pathway exposes only a weak implicit edit signal.

These changes clarify why reasoning improves cross-modal transfer. Under direct generation, Ovis-U1 starts from one of the weakest effective conditioning shifts, consistent with the projection bottleneck discussed above. After reasoning augmentation, however, it becomes the strongest among the evaluated models: its post-reasoning conditioning distance is about 1.67 times that of OmniGen2 and about 2.41 times that of BLIP3o-4B. This reversal suggests that reasoning does not simply add more tokens to the prompt. Rather, it changes the form of the signal delivered to the image generator: the edited model first verbalizes the updated fact, and this explicit textual constraint is then transmitted through the model's ordinary text-to-image conditioning pathway (Zou et al., 2025).

The smaller increase for BLIP3o-4B suggests a different regime. Its direct pathway already exposes more of the edit-induced perturbation, leaving less room for reasoning to further amplify the conditioning shift. In addition, its fixed set of learned query tokens compresses the textual context into a bounded conditioning representation. Ovis-U1, by contrast, conditions on the full text sequence; once the edited knowledge is verbalized, the reasoning trace can induce a much larger semantic shift at the DiT input. This explains why reasoning augmentation is especially effective for Ovis-U1 despite its weak direct edit drift.

## 5. Conclusion

We study a fundamental yet underexplored question in UMMs: do text-side knowledge edits reliably translate into visually verifiable changes in image generation? To answer it, we introduce UNIKE, a cross-modality benchmark that couples textual knowledge edits with VQA-based visual verification. Our results reveal a clear modality gap: editing methods that succeed under text-based metrics do not necessarily induce corresponding visual evidence in generated images. Reasoning-augmented generation partially mitigates this gap by explicitly activating edited knowledge before visual synthesis. Mechanistically, our analysis suggests that this limitation arises from a conditioning-pathway bottleneck: edit-induced changes in textual representations are only partially preserved in the signals used for image generation. Even when the architecture transmits larger conditioning shifts, these shifts remain insufficient to guarantee visually consistent generation. These findings show that textual knowledge edits alone are insufficient for robust cross-modality transfer and motivate future editing methods that directly target modality-relevant generation pathways.

## Acknowledgement

This work is funded in part by the Schmidt Foundation and by the National Science Foundation under grant 2146151.

## Impact Statement

This work advances the field of Machine Learning by investigating knowledge editing in unified multimodal models. The ability to efficiently update factual knowledge without retraining could enable timely corrections of misinformation and customization for specific applications. However, our findings reveal that current editing methods exhibit inconsistent cross-modality transfer, where text-based edits may not reliably propagate to visual generation. This inconsistency could potentially be exploited to create models that verbally claim certain facts while generating contradictory visual content. Additionally, knowledge editing techniques raise concerns about malicious manipulation, such as altering historical facts or creating biased representations. We emphasize that such techniques should be deployed with appropriate safeguards and transparency about modifications.

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

# A. Knowledge Editing Introduction

Knowledge editing asks a simple but practical question: given a pretrained model, how can we *directly rewrite* a specific piece of knowledge in its parameters without retraining the whole model? Let the base model be parameterized by $\theta$, and let $k$ denote the knowledge (e.g., a fact or relation) we want to insert, modify, or erase. An editing algorithm can be viewed as an operator $F$ that produces an updated model

$$\theta' = F(\theta, k). \tag{8}$$

A good edit should change the model's behavior on queries about $k$ (so that the new belief is expressed consistently), while leaving other behaviors as intact as possible. If we use $\theta(k)$ to denote the model's responses (or probabilities) on prompts that test $k$, then the intended effect is

$$\theta'(k) \neq \theta(k), \qquad \text{and ideally} \qquad \forall k' \neq k, \ \theta'(k') \approx \theta(k'). \tag{9}$$

In practice, this objective is usually tested through three complementary lenses: *edit success* (does the model follow the new knowledge on direct prompts?), *generalization* (does it hold under paraphrases or related contexts?), and *locality* (does unrelated knowledge remain stable?). These guarantees are non-trivial because model knowledge is typically *distributed* and *entangled*: the same parameters can support many facts, so even a targeted update may unintentionally perturb other behaviors. Many modern methods therefore adopt a "location-then-edit" style approach: they first identify which internal components contribute most to expressing $k$, and then apply a constrained, parameter-efficient update to those components to make the change persistent in the weights (Zhang et al., 2024a).

A subtle but important distinction is how edits are *evaluated*. A common protocol is the **single-edit** setting, where each test case is edited and evaluated in isolation: for every knowledge item, we apply $F$ to a fresh copy of the original model, evaluate, and then reset/reload the base model before the next item. This protocol makes it easy to attribute outcomes to one edit, but it does not reflect deployment patterns where models receive updates continuously. This motivates **sequential editing** (also called *continuous* editing), where a stream of edits is applied to the *same* model instance:

$$\theta_1 = F(\theta_0, k_1), \quad \theta_2 = F(\theta_1, k_2), \ \dots, \ \theta_T = F(\theta_{T-1}, k_T). \tag{10}$$

Sequential editing is harder for reasons that are easy to miss in single-edit tests: even small off-target changes can *accumulate* across many updates; later edits can *interfere* with or partially overwrite earlier edits; and the method must balance *plasticity* (quickly incorporating new knowledge) against *stability* (preserving both the pre-existing knowledge base and previously applied edits). In other words, single-edit evaluation asks whether an editor can rewrite one fact in isolation, whereas sequential editing asks whether the same editor behaves like a reliable *update mechanism* under repeated edits on a deployed model.

# B. Dataset Introduction

In this section, we briefly introduce some commonly used knowledge editing datasets.

**ZsRE** (Levy et al., 2017) is a question answering dataset that is widely used as a benchmark for evaluating knowledge editing in language models. Under the common editing protocol, each instance specifies a subject entity together with a target answer that serves as the knowledge to be updated, and provides rephrased questions to test whether the effect of an edit persists under semantically equivalent inputs. ZsRE-based evaluations also typically include additional questions that are unrelated to the edited knowledge in order to assess locality and specificity, namely whether the model's behavior on other facts remains stable after applying an edit.

**CounterFact** (Meng et al., 2022) is a benchmark designed to evaluate factual rewriting under stricter locality constraints. Each example pairs an original factual association with a counterfactual target that replaces the original object while keeping the subject and relation fixed, and further provides challenging locality probes by substituting the subject with a semantically similar entity under the same predicate. This design makes CounterFact particularly effective at detecting spillover effects, where an edit intended for one fact inadvertently alters the model's behavior on closely related facts.

**MQuAKE** (Zhong et al., 2023) extends standard single-fact editing evaluations by emphasizing multi-hop reasoning consequences. In addition to an edit target, MQuAKE provides questions whose correct answers depend on the edited knowledge through intermediate reasoning steps, thereby measuring whether the update is reflected in entailed beliefs rather than only in direct recall. As a result, MQuAKE is commonly used to characterize the extent to which editing methods support compositional reasoning consistency after a factual update.

**TMKE** (Fang et al., 2025b) is a benchmark that evaluates the transitivity of knowledge editing across modalities in vision-language models. It formalizes settings in which a knowledge update is applied in one modality and is subsequently tested under multimodal inputs, and it measures whether the updated belief is expressed consistently across different prompt formulations and visual contexts. TMKE therefore provides a structured framework for assessing cross-modality generalization and robustness of edited knowledge in multimodal systems.

## C. Softened AlphaEdit Projector

AlphaEdit (Fang et al., 2025a) constrains parameter updates to directions that have low activation energy under the pre-edit model, thereby reducing interference with previously learned behavior. For a layer $l$, AlphaEdit estimates an empirical second-moment matrix from pre-edit key activations,

$$C_l = \frac{1}{N} \sum_{i=1}^{N} k_i k_i^\top, \tag{11}$$

and computes its eigendecomposition $C_l = U \Lambda U^\top$. A null-space projector is then constructed as

$$P_l = U_\mathcal{N} U_\mathcal{N}^\top, \tag{12}$$

where $U_\mathcal{N}$ contains the eigenvectors whose eigenvalues are below a threshold $\tau$. The resulting update is restricted to the range of $P_l$, so that it primarily modifies directions that are weakly used by the pre-edit activation distribution.

This constraint is effective when there exists a sufficiently large low-energy subspace for editing. However, in UMMs with shared visual backbones, the same transformer layers participate in both textual processing and visual conditioning. This is the case for BLIP3o-4B and OmniGen2, where the Qwen2.5-VL backbone is shared across modalities. As a result, the activation covariance can occupy a larger portion of the representation space, leaving a smaller effective null space. Applying the original AlphaEdit projector in this setting can therefore make the update overly restrictive, limiting the editor's ability to encode the target knowledge.

**Softened projector.**    To relax this constraint, we use a softened projector that interpolates between the AlphaEdit null-space projector and the identity matrix:

$$\tilde{P}_l = \alpha P_l + (1 - \alpha) I, \tag{13}$$

where $\alpha \in [0, 1]$ controls the strength of null-space enforcement. When $\alpha = 1$, the method reduces to the original AlphaEdit projector. When $\alpha = 0$, the update is unconstrained by the null-space projection. Intermediate values preserve the locality bias of AlphaEdit while allowing part of the update to enter higher-energy directions when the null space is insufficient for effective editing.

Using this softened projector, the layer update is computed as

$$\Delta W_l = \left( \tilde{P}_l (K_l K_l^\top + C_{\text{cache}}) + \lambda I \right)^{-1} \tilde{P}_l K_l R_l^\top, \tag{14}$$

where $K_l$ denotes the key activations for the current edit batch, $R_l$ is the residual target distributed to layer $l$, $C_{\text{cache}}$ accumulates key outer products from previous sequential edits, and $\lambda$ is an $L_2$ regularization coefficient.

**Covariance regularization.**    For shared visual backbones, we also regularize the second-moment matrix before eigendecomposition:

$$C_l \leftarrow C_l + \epsilon I. \tag{15}$$

This prevents near-zero eigenvalues caused by finite-sample estimation from being treated as reliable null directions. In practice, this yields a more stable projector and avoids overly aggressive updates along poorly estimated directions.

**Configuration.**    We apply the softened projector only to models whose language backbone is shared with the visual-conditioning pathway. Specifically, we use

- **BLIP3o-4B**: $\alpha = 0.7$, $\epsilon = 0.01$;

- **OmniGen2**: $\alpha = 0.6$, $\epsilon = 0.015$;

- **Ovis-U1**: standard AlphaEdit with $\alpha = 1.0$ and no additional covariance regularization.

For BLIP3o-4B and OmniGen2, we additionally clip the update norm to avoid abrupt changes in the shared backbone. The maximum update ratios are set to $0.5$ and $0.4$, respectively, where the ratio is measured as $\|\Delta W_l\|_F / \|W_l\|_F$. These hyperparameters were chosen to balance text-side edit efficacy with stable image-generation behavior under sequential editing.

## D. Conditioning Pathway Analysis

This section provides detailed experimental results supporting the mechanistic analysis in Section 4.6. We first summarize the conditioning architecture of each model, then present the SVD bottleneck analysis for Ovis-U1, followed by the full conditioning drift breakdowns, DiT propagation verification, and reasoning decomposition results.

### D.1. Conditional architecture Difference

Ovis-U1 concatenates the last two language hidden layers and maps them through a frozen linear projection before passing them to the DiT. BLIP3o-4B appends 64 learnable query tokens to the text sequence; these queries pass through the same edited LLM layers, and their hidden states are used directly as DiT conditioning. OmniGen2 passes final-layer text hidden states directly to the DiT. The differences are summarized to Table 5.

*Table 5.* Conditioning pathway architectures of the three evaluated models.

| Property | Ovis-U1 | BLIP3o-4B | OmniGen2 |
|---|---|---|---|
| LLM backbone | Qwen3-1.7B | Qwen2.5-VL | Qwen2.5-VL |
| Edited layers | [4, 5, 6, 7, 8] | [6, 7, 8, 9, 10] | [6, 7, 8, 9, 10] |
| Conditioning dimension | 4096 (concat 2 layers) | 2048 | 2048 |
| N conditioning tokens | All text tokens | 64 (learned queries) | All text tokens |
| Projection to DiT | Linear(4096, 1536), frozen | None | None |

### D.2. SVD Bottleneck Analysis

Ovis-U1's conditioning pathway contains a frozen linear projection $W \in \mathbb{R}^{1536 \times 4096}$ that maps the concatenated last-two-layer hidden states into the DiT's input space. To quantify how much edit signal this projection preserves (Sharma et al., 2024), we perform singular value decomposition $W = U\Sigma V^\top$ and measure the cumulative fraction of the edit perturbation captured by the top-$k$ right singular vectors:

$$\rho(k) = \frac{\|V_{:k}^\top \delta\|^2}{\|\delta\|^2}, \tag{16}$$

where $V_{:k}$ denotes the top-$k$ right singular vectors, which span the dominant input directions preserved by the projection, and $\delta = c_{\text{edit}} - c_{\text{fresh}}$ is the edit perturbation in the 4096-dimensional pre-projection space.

Table 6 reports $\rho(k)$ for Ovis-U1 with PMET on 100 edit cases. The near-linear growth suggests that the observed edit perturbations are broadly distributed across the 4096-dimensional space, with no strong preferential alignment to the projection's principal directions. This matches the theoretical expectation: for a uniformly distributed vector in $\mathbb{R}^n$, a linear projection to $\mathbb{R}^k$ preserves $k/n$ of its squared norm in expectation, yielding $1536/4096 = 37.5\%$ for this setting. This indicates that, for the current MLP-based editors and their observed perturbation patterns, the frozen projection imposes a fixed retention bottleneck unless the edit signal is explicitly aligned with the projection-preserved subspace or the projection itself is modified.

Table 7 demonstrates that the signal retention ratio remains stable at 32–35% across all three editing methods, establishing this as a fixed architectural property independent of the editing algorithm.

### D.3. DiT Block Sensitivity

To verify that the performance gap arises at the conditioning interface rather than within the DiT, we trace the edit-induced perturbation through all transformer blocks of Ovis-U1's Yak DiT at timestep $t=0.5$. We use identical random image latents

*Table 6.* Cumulative fraction $\rho(k)$ of edit-perturbation energy captured by the top-$k$ right-singular input directions of Ovis-U1's frozen projection, computed from 100 edit cases edited with PMET.

| Top-$k$ | $\rho(k)$ | Theoretical $k/4096$ |
|---|---|---|
| 1 | 0.01% | 0.02% |
| 10 | 0.18% | 0.24% |
| 100 | 1.99% | 2.44% |
| 500 | 10.62% | 12.21% |
| 1000 | 21.60% | 24.41% |
| 1536 | 34.82% | 37.50% |

*Table 7.* Signal retention through Ovis-U1's frozen projection across editing methods (100 cases each). $\|\delta\|$: pre-projection perturbation norm. $\|W\delta\|$: post-projection perturbation norm. $\rho(1536)$: fraction of perturbation energy in the row space of the projection.

| Method | $\|\delta\|$ | $\|W\delta\|$ | $\rho(1536)$ |
|---|---|---|---|
| AlphaEdit | 2810.5 | 585.0 | 34.2% |
| MEMIT | 4134.8 | 973.3 | 31.9% |
| PMET | 2930.6 | 669.6 | 34.8% |

for both fresh and edited models to isolate the effect of conditioning differences. As shown in Table 8, the drift magnitude remains within 2% variation across all single-stream blocks, confirming that the DiT operates in a linear regime at this perturbation magnitude and neither amplifies nor attenuates the signal.

The relative drift at Ovis-U1's DiT input is 0.077, compared to 0.458 for BLIP3o-4B and 0.435 for OmniGen2 (measured at the conditioning interface). This 5–6$\times$ gap at the DiT input directly constrains achievable image generation differences, confirming that the bottleneck resides upstream at the projection interface.

*Table 8.* Per-block DiT drift for Ovis-U1 (AlphaEdit, 100 cases). The near-constant drift across blocks confirms linear propagation without amplification.

| Stage | Absolute Drift | Relative to Block 0 |
|---|---|---|
| txt_in projection | 7,273 | 1.000 |
| SingleStreamBlock 0 | 7,243 | 0.996 |
| SingleStreamBlock 1 | 7,238 | 0.995 |
| SingleStreamBlock 2 | 7,187 | 0.988 |
| SingleStreamBlock 3 | 7,225 | 0.993 |
| SingleStreamBlock 4 | 7,398 | 1.017 |

# E. Case Study

In this section, we present a representative qualitative example under the *reasoning-augmented* editing protocol, illustrating how the reasoning chain mediates cross-modality knowledge transfer.

Figure 6 shows a multi-stage *pattern* edit on Ovis-U1 with PMET, where the peacock's plumage pattern is edited from "iridescent eyed" to "floral." In the reasoning stage, the edited model consistently recalls the target attribute across all generation stages: it identifies the surface pattern as floral, describes its physical manifestation (flowers, leaves, and vines), and specifies how the pattern interacts with each scene's environment. This structured reasoning output then serves as the conditioning text for image generation.

The pre-edit model generates peacocks with the standard eye-spot plumage pattern regardless of scene context. After editing, the reasoning-augmented protocol produces images where floral motifs are visible on the peacock's body (Stages 1–2) and on a derived object (Stage 4, a peacock-inspired scarf). The success across stages demonstrates that when reasoning correctly verbalizes the edited knowledge, the full-sequence text conditioning in Ovis-U1 can translate it into consistent visual changes—even for fine-grained surface attributes that require spatially coherent rendering.

**Figure 6.** Qualitative examples of reasoning-augmented cross-modality knowledge editing. Each row shows a different generation stage for the same edit (Peacock pattern: "iridescent eyed" → "floral"). Left: metadata and image prompt. Middle-left: pre-edit generation. Middle-right: structured reasoning output from the edited model. Right: post-edit generation. The reasoning chain consistently recalls the edited attribute ("floral") and produces a visual description grounded in the target pattern, which is then reflected in the generated images across progressively more complex scenes.

## F. Prompt

For data generation (Appendix F.1 and F.2), we use Gemini-3.0-Flash with temperature 1.0 to encourage diversity in entity coverage, attribute selection, and prompt formulation. For LMM-based visual verification (Appendix F.3), we use Qwen3-VL-235B-A22B-Instruct with temperature 0.1 to ensure deterministic and consistent judgment (Shi et al., 2024).

### F.1. Prompt for generating attribute data

**Attribute Category Descriptions**

```
#### Color
Focuses on changing the visual hue or color of an object.

CRITICAL GUIDELINES:
1. Cloze/Statement Format: Prompts MUST be incomplete statements that the model
    completes.
    USE:   "The color of the tomato is", "The color of the sky is"
```

```
      AVOID: "What color is...?", "How does it look?"
2. Concise Noun Answers: gt and gt\_target must be the COLOR NAME ONLY (1-2 words).
   BAD:  "The tomato has a bright blue skin."
   GOOD: "blue"

Examples of Good Color Edits:
- "The color of this tomato is" (GT: "red" -> Target: "blue")
- "The color of the leaves on this tree is" (GT: "green" -> Target: "pink")
- "The color of the ocean water is" (GT: "blue" -> Target: "red")

#### Material
Focuses on changing the fundamental substance or composition of an object.

CRITICAL GUIDELINES:
1. Cloze/Statement Format: Prompts MUST be incomplete statements that the model
    completes.
   USE:   "The violin is made of", "The cloud is composed of"
   AVOID: "What material is...?", "What happens if...", "How does it interact..."
2. Concise Noun Answers: gt and gt\_target must be the MATERIAL NAME ONLY (1-2
    words).
   BAD:  "The light passes through the transparent glass."
   GOOD: "glass"

Examples of Good Material Edits:
- "The teddy bear is made of" (GT: "fur" -> Target: "metal")
- "The cloud in the sky is composed of" (GT: "vapor" -> Target: "concrete")
- "The violin is constructed from" (GT: "wood" -> Target: "glass")
- "The statue is made of" (GT: "stone" -> Target: "water")

#### Shape
Focuses on changing the geometric 3D form or topology of an object.

CRITICAL GUIDELINES:
1. Cloze/Statement Format: Prompts MUST be incomplete statements that the model
    completes.
   USE:   "The shape of the watermelon is", "The geometric form of the ball is"
   AVOID: "What shape is...?", "Describe the structure..."
2. Concise Noun Answers: gt and gt\_target must be the SHAPE NAME ONLY (1-2 words).
   BAD:  "It is shaped like a perfect cube."
   GOOD: "cube"

Examples of Good Shape Edits:
- "The geometric shape of this watermelon is" (GT: "oval" -> Target: "cube")
- "The shape of this soccer ball is" (GT: "sphere" -> Target: "pyramid")
- "The structural shape of the Earth is" (GT: "sphere" -> Target: "flat")

#### Size
Focuses on changing the relative size of an object.

CRITICAL GUIDELINES:

1. Stage 1: Absolute Size Change
- Prompt: Use statement format about general physical size/scale.
  USE: "The typical size of an ant is"
- GT / Target: Use adjectives describing the scale.
  GT: "tiny" / "small" / "microscopic"
  Target: "colossal" / "giant" / "enormous" / "building-sized"

2. Stages 2-4: Direct Comparisons
- Use Reference Objects: Always compare the entity to a familiar object.
- Comparison Statements: Use statements where the answer is the larger/smaller
    object.
```

```
   USE: "Between the ant and the shoe, the larger one is"
   AVOID: "Which is larger...?" (question format)
- Concise Answers (1-2 Words): GT and GT\_TARGET must be ONLY the name of the object
   (e.g., "shoe", "ant"). NO explanations.

Examples:
- Direct: "The size of an ant is" (GT: "tiny" -> Target: "colossal")
- Contextual: "Between the ant and the fire hydrant, the taller one is"
   (GT: "fire hydrant" -> Target: "ant")

Image Prompts: MUST include a reference object in ALL stages.

#### Pattern
Focuses on changing the surface design or print of an object.

CRITICAL GUIDELINES:
1. Cloze/Statement Format: Prompts MUST be incomplete statements that the model
   completes.
   USE:  "The pattern on the zebra is", "The design on the shirt is"
   AVOID: "What pattern is...?", "Describe the markings..."
2. Concise Noun Answers: gt and gt\_target must be the PATTERN NAME ONLY (1-2 words).
   BAD:  "It has black and white stripes."
   GOOD: "stripes" / "polka dots"

Examples of Good Pattern Edits:
- "The pattern on the zebra's coat is" (GT: "stripes" -> Target: "dots")
- "The design pattern on the snake's skin is" (GT: "scales" -> Target: "checkered")
- "The pattern on the surface of the apple is" (GT: "solid" -> Target: "plaid")
```

## Prompt Template for Stage Generation

```
You are helping to construct a high-quality dataset of multi-stage, multi-modal
   knowledge editing examples.
Your task is to generate ONE high-quality example for the specific entity provided
   below.

---

Target Entity:
Entity: \{entity\}
\{default\_attribute\_line\}

---

Your Task:

1. Use the specific entity provided above.
2. Choose one attribute to edit within the specified category.
3. Specify the original value (gt) (should match the provided default if applicable)
    and a distinct, counterfactual edited value (gt\_target).
4. Generate prompts and image prompts for 4 stages of increasing complexity.
5. CRITICAL: Eliminate ambiguity. Do NOT use pronouns (e.g., "this banana", "it",
    "the bird", "that car").
   Always repeat the full entity name (e.g., "the banana", "the flamingo") in every
    single prompt and image prompt.
6. IMPORTANT: In the prompt, do NOT mention the edited value.

---

Prompt Format (CRITICAL - Cloze/Statement Style):
```

```
ALL prompts MUST be incomplete statements that the model completes, NOT questions.

CORRECT FORMAT:
- "The color of the banana is"
- "The material of the violin is"
- "The shape of the watermelon is"

WRONG FORMAT (DO NOT USE):
- "What color is the banana?"
- "What material is the violin made of?"
- "What is the shape of the watermelon?"

---

Stages Definition:

Stage 1: Direct reference to the edited attribute
- Prompt: State directly about the attribute.
- Image prompt: Show the entity (NEUTRAL - don't mention edited value).

Stage 2: The entity appears in a realistic scenario or context
- Prompt: State about the attribute in this context.
- Image prompt: Show a natural scene (NEUTRAL - don't mention edited value).

Stage 3: Complex scenes with multiple entities and interactions
- Prompt: State about the attribute despite the complexity.
- Image prompt: Show a rich, detailed scene (NEUTRAL - don't mention edited value).

Stage 4: Products, derivatives, or direct uses that inherit the attribute
- Prompt: State about the attribute of the derived product.
- Image prompt: Show the derived product (NEUTRAL - don't mention edited value).

IMPORTANT: In all stages, gt and gt\_target should be CONCISE (1-2 words). NO
    explanations.
The prompt should be a statement that naturally leads to a 1-2 word answer.

---

Image Prompt Rules (CRITICAL - Answer Neutrality):

Image prompts are fed to a Text-to-Image model. The rendered image is then judged
    against visual\_target. The image\_prompt must NOT reveal the answer.

1. Must describe a concrete scene that physically contains the object whose
    attribute is under test (for stage\_4, that may be a derivative, e.g. ``tomato
    sauce'', ``orange juice'').
2. Must name the subject explicitly (or the named derivative for stage\_4).
3. Must NOT mention the edited value (gt\_target) literally.
4. Must NOT mention the original value (gt) literally.
5. Must NOT describe the visual appearance implied by either value (e.g. for
    ``Tomato: red -> blue'', do NOT write "the tomato glistening with
    sapphire-coloured flesh"; just write "a close-up photo of a tomato sliced on a
    wooden cutting board").

---

VQA Question and Visual Target (For VLM Judge Evaluation):

Each stage also requires:
1) vqa\_question: a question-format version of the prompt for a VLM Judge.
```

```
2) visual\_target: a strict declarative statement asserting the subject in the image
    MUST have the target property.

VQA Question Rules:
- MUST contain the subject name explicitly (e.g. "What is the color of the
    strawberry in this image?"). For stage\_4 (derived), must contain the explicit
    derived object noun (e.g. "the tomato sauce", "the orange juice").
- Do NOT replace the subject with a generic descriptor like "the fruit", "the
    animal", "the object".
- MUST NOT contain the gt or gt\_target strings (size category is exempt: a
    comparative question may legitimately name both comparison objects).
- Must be answerable from the image alone, with the correct answer = gt\_target.

Visual Target Rules:
- Must be a single declarative sentence asserting the attribute visible in the scene.
- For color/material/shape/pattern: assert a property of the actual object shown,
    e.g. "The tomato sauce in the bowl must be blue."
- For SIZE (special rule): the visual\_target MUST assert a relative size relation,
    not an absolute adjective. Use templates like: "The \{subject\} in the image must
    appear much larger than \{reference\}."
- The object referenced in visual\_target MUST actually appear in image\_prompt.
- Must NEVER be a tautology ("the X must be an X").

Examples:
| prompt | vqa\_question | visual\_target |
| "The color of the banana is" | "What color is the banana in this image?" | "The
    banana in this image must be green." |
| "The material of the violin is" | "What material is the violin made of in this
    image?" | "The violin in this image must be made of glass." |
| "The shape of the watermelon is" | "What is the shape of the watermelon in this
    image?" | "The watermelon in this image must be cube-shaped." |

---

Critical Requirements:

1. Distinct/Counterfactual Edits
- gt\_target must be clearly distinct from gt and represent a counterfactual change.

2. Neutral Image Prompts (NO LEAKAGE)
- Image prompts must NOT explicitly mention the edited value or the original value.
- Must NOT describe the visual appearance implied by either value.

3. NO AMBIGUITY (No Pronouns)
- Every prompt must be self-contained and repeat the full entity name.

4. Visual Testability
- Each stage must produce a visually distinguishable outcome. If the (subject,
    gt\_target) pair cannot be rendered as a distinguishable image for a given stage,
    mark that stage as unsuitable.

---

Output Format:

Return a single JSON object (or a list containing one object) in the following
    format:

[
  \{
    "entity": "\{entity\}",
    "attribute": "attribute\_name",
```

```
      "gt": "original\_value",
      "gt\_target": "edited\_value",
      "stage\_1": \{
        "prompt": "The [attribute] of the [entity] is",
        "gt": "...",
        "gt\_target": "...",
        "image\_prompt": "...",
        "visual\_target": "...",
        "vqa\_question": "What [attribute] is the [entity] in this image?"
      \},
      "stage\_2": \{ ... \},
      "stage\_3": \{ ... \},
      "stage\_4": \{ ... \}
  \}
]

---

Category Focus:

\{insert\_category\_description\}

---

Constraints:
- Generate exactly ONE example for the entity "\{entity\}".
- Output must be pure JSON.
```

## F.2. Prompt for generating relation data

---

**Prompt for Filtering and Specifying Visualizable Knowledge Facts**

```
You are helping to construct a cross-modal knowledge-editing benchmark for
    relational edits. For ONE fact, we need to (a) decide whether the counter-factual
    edit is testable via an image, and (b) if so produce image\_prompt /
    visual\_target / vqa\_question fields under strict rules.

=== FACT ===
- Subject: \{subject\}
- Question: \{question\}
- Prompt template: \{prompt\}
- Original answer (gt): \{gt\}
- Edited answer (gt\_target): \{gt\_target\}

=== DECIDE SUITABILITY ===
Return suitable: false for any of the following (with a short skip\_reason):
- The test hinges purely on recognising the face of a specific person whom only
    large models are likely to know.
- Denomination-level religion edits that are not visually distinguishable (e.g.
    Catholic vs Church of Scotland).
- Abstract / etymological relations: named after, called after, native to,
    citizenship, mother tongue, language of, died in, born in, founded in, treaties,
    twin cities.
- gt\_target that is semantically incompatible with the subject to the point that no
    coherent image is possible.
- Spouses / children of specific named humans (the edited person needs to be
    rendered AND recognised).
- When gt is a substring of subject (the original answer is baked into the name).
```

```
Otherwise set suitable: true and proceed.

=== RULES FOR GENERATED FIELDS ===
1. image\_prompt: CRITICAL --- NEUTRAL, DO NOT LEAK THE EDITED ANSWER.

   This prompt is fed to a Text-to-Image model. The rendered image is then judged
    against visual\_target. The whole point of the benchmark is to verify that the
    edited model produces an image consistent with gt\_target even though
    image\_prompt does NOT tell it the answer.

   Therefore the image\_prompt must NOT describe ANY visual element that uniquely
    belongs to either the original answer (gt) or the edited answer (gt\_target). It
    must only set up the stage where the answer would be visible if the model knew it.

   GOOD (NEUTRAL --- describes a stage where the answer is implicit):
     * "A professional photo of \{subject\} at his workplace engaged in his domain
    of activity."
     * "A photo of \{subject\} doing his job."
     * "A close-up photo of the \{subject\} motorcycle clearly showing its
    manufacturer's logo on the fuel tank."

   BAD (LEAKS --- describes the visual context of gt or gt\_target):
     * "A portrait of Eric Maskin in a clinical setting wearing a white lab coat
    with a stethoscope."
     * "A photo of \{subject\} writing poetry at his desk."
     * "A wide-angle street view with distinctive South-American architecture."

   Hard MUST-NOT rules:
   - MUST NOT mention gt or gt\_target literally (string-level).
   - MUST NOT name any attribute that uniquely identifies gt or gt\_target.
   - MUST NOT exceed approximately 25 words.

2. visual\_target\_role (primary visual check):
   - Verifiable by a small VLM (no celebrity-face recognition required).
   - One declarative sentence that asserts what role/attribute the scene must show,
    using gt\_target.
   - Examples:
     * "The logo on the motorcycle must read 'Porsche'."
     * "The architecture, signage and landscape in the image must clearly belong to
    Ukraine."
     * "The person in the image must be dressed in the uniform of a linebacker."

3. visual\_target\_identity (optional secondary target):
   - Include only when the identity can plausibly be rendered AND recognised.
   - One declarative sentence, e.g. "The person in the portrait must be identifiable
    as Humza Yousaf."
   - Set to null when identity recognition is not feasible.

4. vqa\_question:
   - MUST NOT contain the subject string.
   - MUST NOT contain gt or gt\_target.
   - Refers to the entity by its role in the image ("the person in the portrait",
    "the car in the photo").
   - The correct answer must be gt\_target.

5. category:
   - Pick ONE: affiliation, creator, location, and occupation.
   - Items classified as "other" are dropped downstream.

=== OUTPUT (return ONLY this JSON object) ===
\{
```

```
  "suitable": bool,
  "skip\_reason": str|null,
  "category": str,
  "image\_prompt": str|null,
  "visual\_target\_role": str|null,
  "visual\_target\_identity": str|null,
  "vqa\_question": str|null
\}
```

### F.3. Prompt for VLM Judge

**Prompt for VQA-based Visual Verification**

```
Look at this image carefully and answer the following question.

Question: \{vqa\_question\}

Target Criterion (The image MUST satisfy this statement): \{visual\_target\}

Your task:
1. Describe what you observe in the image related to the question.
2. Determine if the image STRICTLY satisfies the Target Criterion.
   - If the Target Criterion says "must be X", and the image shows Y, then it does
    NOT match.
   - Be rigorous. The image must clearly demonstrate the target property.
3. Respond ONLY with a JSON object in this exact format:
\{
    "observation": "describe what you see in the image",
    "matches\_target": true or false,
    "confidence": "high", "medium", or "low",
    "explanation": "why you think it matches or doesn't match the expected target"
\}

Respond ONLY with the JSON object, no other text.
```

