# OpenReview forum: "Do Text Edits Generalize to Visual Generation? Benchmarking Cross-Modal Knowledge Editing in UMMs"
_ICML.cc/2026/Conference — ICML 2026 regular_

### Official Review · Reviewer_Mceo · 2026-03-11

**Soundness:** 3
**Presentation:** 2
**Significance:** 2
**Originality:** 3
**Overall Recommendation:** 4
**Confidence:** 3

**Summary:**

This paper investigates whether text-side knowledge edits in UMMs transfer to image generation. The authors introduce UNIKE, the first benchmark for cross-modality knowledge editing in UMMs, comprising 3,005 instances across attribute edits  and relation edits. An automated VQA-based evaluation protocol using Qwen3-VL-235B as a judge is proposed. Experiments on Ovis-U1 reveal a striking modality gap: parameter-editing methods achieving up to 93% text-side efficacy yield below 6% VQA accuracy under direct generation. The authors propose Reasoning-augmented Parameter Editing, which explicitly activates edited knowledge before generation, improving VQA accuracy to 10–27% for attributes. A mechanistic analysis shows that edit-affected channels exhibit near-random overlap with visual attribute-conditioning channels, providing an interpretable explanation for the cross-modal transfer failure.

**Compliance With Llm Reviewing Policy:**

Affirmed.

**Final Justification:**

The reviewer resolved some of my doubts. I improved my rating.

**Key Questions For Authors:**

1、Do the authors plan to replicate experiments on other UMMs with different architectures?
2、Given that attribute-sensitive channels (set B(K)) have been identified, have the authors attempted to directly edit these channels instead of relying on standard KE methods that target different pathways? Even a preliminary modality-aware editing experiment would substantially strengthen the paper's contribution.

**Limitations:**

Yes

**Strengths And Weaknesses:**

Strengths:
1、The research question is novel and forward-looking. With the rise of UMMs such as Gemini, Emu3.5, and Ovis-U1, cross-modality knowledge editing is a natural and important direction. This is the first systematic study in this space, and UNIKE has the potential to serve as a standard testbed for future work.
2、The benchmark construction pipeline is rigorous. The pre-edit success filter ensures that only existing model knowledge is targeted, and the visualizability constraint ensures that edit targets admit verifiable visual implications. The four-stage progressive design for attribute edits effectively quantifies difficulty, and the stage-wise analysis in Figure 4 validates this design.
3、The pathway mismatch analysis is the valuable contribution. Using Jaccard similarity under a top-K sweep, the authors quantitatively show that the overlap between edit-affected channels and attribute-sensitive channels is indistinguishable from a random baseline. This finding offers direct guidance for designing future modality-aware editing methods.

Weekness:
1、All experiments are conducted on a single UMM (Ovis-U1 For a benchmark and analysis paper, single-model conclusions raise generalizability concerns. Multiple UMMs with different architecturesshould be evaluated to confirm whether pathway mismatch is a universal property of UMMs or an artifact of the Ovis-U1 architecture.
2、The pathway mismatch analysis uses only 60 prompt pairs for the color attribute，the easiest category according to Figure 5. It is unclear whether the same analysis holds for harder attributesand relation categories.
3、The proposed Reasoning-augmented Parameter Editing yields modest improvements at best VQA accuracy reaches only 16–27% for attributes and 6–7% for relations. The paper primarily identifies the problem rather than solving it.

---

> ### Author Rebuttal · Authors · 2026-03-31
>
> # Response 1
>
> We thank the reviewer for this important suggestion. In response, we have extended our experiments to two additional UMMs with distinct architectures: BLIP3o-4B [1] and OmniGen2 [2]. We applied the same knowledge editing methods to these models using the same benchmark protocol. Across all three architectures, the direct cross-modal transfer remains consistently low, with VQA accuracy for direct generation below 10%, reinforcing that the modality gap is a shared architectural property of UMMs rather than a model-specific artifact. As also noted in our response to Reviewer 1 (Response 2), the interpretability findings are equally consistent. The full experimental results tables will be provided in the additional comments.
>
> # Response 2
>
> We thank the reviewer for this insightful comment. We agree that restricting the pathway mismatch analysis to color alone is insufficient. We have now extended the analysis to three attribute categories (color, shape, and affiliation) across three model backbones (Ovis-U1-3B, BLIP3o-4B, and OmniGen2). For each category, we construct 60 matched prompt pairs following the same methodology, where each pair differs only in an attribute term, for example, "A round clock" vs. "A clock" for shape; "A Catholic cathedral" vs. "A cathedral" for affiliation. The Jaccard overlap J(K) is averaged over 10 single-edit trials per category.
>
> | K | Ovis/color | Ovis/shape | Ovis/affil. | BLIP3o/color | BLIP3o/shape | BLIP3o/affil. | OmniGen2/color | OmniGen2/shape | OmniGen2/affil. | Random |
> |---:|---:|---:|---:|---:|---:|---:|---:|---:|---:|---:|
> | 10 | 0.0132 | 0.0010 | 0.0081 | 0.0123 | 0.0206 | 0.0133 | 0.0102 | 0.0227 | 0.0166 | 0.0028 |
> | 20 | 0.0122 | 0.0076 | 0.0066 | 0.0112 | 0.0173 | 0.0133 | 0.0117 | 0.0174 | 0.0133 | 0.0046 |
> | 50 | 0.0155 | 0.0132 | 0.0122 | 0.0138 | 0.0185 | 0.0175 | 0.0155 | 0.0203 | 0.0171 | 0.0129 |
> | 100 | 0.0266 | 0.0227 | 0.0224 | 0.0255 | 0.0288 | 0.0281 | 0.0278 | 0.0315 | 0.0266 | 0.0253 |
> | 200 | 0.0516 | 0.0513 | 0.0477 | 0.0511 | 0.0528 | 0.0545 | 0.0547 | 0.0561 | 0.0508 | 0.0514 |
> | 500 | 0.1400 | 0.1388 | 0.1373 | 0.1385 | 0.1376 | 0.1426 | 0.1406 | 0.1396 | 0.1435 | 0.1393 |
> | 1000 | 0.3226 | 0.3229 | 0.3220 | 0.3228 | 0.3220 | 0.3241 | 0.3228 | 0.3221 | 0.3240 | 0.3229 |
>
> Across all nine (model, attribute) combinations, J(K) closely tracks the random baseline for K ≥ 50. Minor elevations at very small K remain within the one-sigma random envelope in absolute terms (J < 0.025) and are not statistically significant. The harder categories, such as the shape and affiliation, exhibit the same near-chance overlap as color, and this pattern is consistent across all three architectures. These results confirm that the pathway mismatch generalizes: the MLP channels perturbed by text-side editing are statistically independent of the channels encoding attribute-specific visual conditioning, regardless of attribute type or model architecture. This consistency across three fundamentally different attribute categories (perceptual color, geometric shape, and semantic affiliation) and three distinct architectures also indicates that 60 prompt pairs per category provide sufficient statistical power to reliably detect the phenomenon.
>
> # Response 3
>
> We thank the reviewer for their careful reading. We wish to emphasize that the primary contribution of our paper is to identify, quantify, and mechanistically explain the cross-modality gap, rather than to propose a fully resolved editing method. Building on the mechanistic findings in Section 4.7, we are currently conducting experiments on modality-aware editing. Concretely, rather than applying standard methods that modify MLP weights in layers identified through causal tracing on text completions, our modality-aware approach identifies and directly targets the attribute-sensitive channel set B(K) that governs visual conditioning. This strategy is motivated by our finding that channels perturbed by standard editing exhibit near-random overlap with visual-conditioning channels (as shown in Response 2), suggesting that standard methods inherently modify the wrong pathways. We will include preliminary results comparing standard and modality-aware approaches in the additional comments and provide a full treatment in the revised manuscript.
>
> [1] BLIP3-o: A Family of Fully Open Unified Multimodal Models
>
> [2] OmniGen2: Exploration to Advanced Multimodal Generation

---

> > ### Author Rebuttal · Reviewer_Mceo · 2026-04-02
> >
> > The reviewer resolved some of my doubts.

---

> > > ### Author Response · Authors · 2026-04-08
> > >
> > > Thank you very much for your valuable suggestions and for your final acknowledgment. We sincerely appreciate your positive feedback and are glad that our rebuttal was able to address your concerns. We are truly grateful for your support and recognition.

---

### Official Review · Reviewer_eLZP · 2026-03-11

**Soundness:** 2
**Presentation:** 3
**Significance:** 3
**Originality:** 3
**Overall Recommendation:** 3
**Confidence:** 3

**Summary:**

This paper proposes a new benchmark that evaluates text-based model edits for unified multi models can be transferred to visual generations. Empirical observations suggest this is not necessarily the case, suggesting considerable room for improvement for future works.

**Compliance With Llm Reviewing Policy:**

Affirmed.

**Final Justification:**

I keep my rating for now and wait for "full comparison results in the additional comments" indicated by authors in Response 1

**Key Questions For Authors:**

Overall, my main concern of the paper is the lack of validation of the benchmark and evaluation metrics.
1. What is the accuracy of Qwen3-VL-235B-A22B-Instruct on the proposed VQA tasks? Is this the best available open-source model for this task? Currently, this choice is not well-justified. The authors are suggested to create a set of human-verified "gold annotations" and evaluate VQA accuracy using a wide range of model choices (e.g. Intern-VL3). Finally, the best model is selected as the judge. This comparison will make the benchmark score more convincing.
2. What is the quality of Gemini-3.0-Flash-based filtering? Ideally, benchmark should be human-verified. While this process can be costly and can be reasonably replaced with LLMs, the author should conduct a careful review to make sure 1. The LLMs can accurately determine if a question is depicable.  2.  The LLMs do not introduce any systematic biases (e.g. treat different regions/races differently, such as consider Paris as "depictable" while consider Cape Town as "not depictable"). This can be done via human audit on a small subset of generated outputs of LLMs.

**Limitations:**

yes

**Strengths And Weaknesses:**

Strength

1. Whether text-based model edits can be transferred to visual generations is a relevant but underexplored topic. This paper offers considerable contribution in this direction.
2. Evaluation results suggest there are considerable room for future works in this direction, which makes the benchmark more impactful.

Weakness
1.  The dataset filtering is done via a LLM, which is known to have hallucinations.
2.  Evaluation metric involving using a VLM to perform VQA tasks, however VLM may produce error. It is also unclear if the selected VLM can perform the specific type of VQA tasks well

---

> ### Author Rebuttal · Authors · 2026-03-31
>
> # Response 1
>
> We thank the reviewer for this important suggestion. We are conducting a comprehensive evaluation of multiple VLMs on our proposed VQA tasks with human-verified gold annotations to justify the model selection. We will provide the full comparison results in the additional comments.
>
> # Response 2
>
> We thank the reviewer for this constructive comment. We agree that the quality of LLM-based filtering should be carefully verified. To address this concern, we conducted a systematic human audit of Gemini 3.0 Flash's depictability judgments on a stratified random sample of 121 examples: 51 items Gemini accepted as depictable (Group A), 50 items Gemini rejected as not depictable (Group B), and 20 location items from diverse geographic regions, including Western and non-Western countries (Group C). All groups were stratified by category and source dataset, and items rejected due to API transient failures were excluded to isolate Gemini's genuine reasoning quality.
>
> **1. Inter-Rater Agreement.** We treat this as an inter-rater reliability problem between Gemini and the human auditor. The contingency table is:
>
> |                    | Human = suitable | Human = unsuitable |
> |--------------------|:----------------:|:------------------:|
> | **Gemini = suitable**   |       52         |         9          |
> | **Gemini = unsuitable** |        4         |        56          |
>
> | Metric | Value | Interpretation |
> |--------|:-----:|:--------------:|
> | Agreement | 89.3% | — |
> | Cohen's κ | 0.79 | Substantial |
> | Krippendorff's α | 0.79 | Substantial |
>
> Both Cohen's κ and Krippendorff's α fall in the "Substantial" range under the Landis and Koch (1977) scale [1], consistent with reliable annotation quality.
>
> **2. Filtering Accuracy.**
>
> Here, we evaluated two different metrics: precision, which is measured by the accepted items truly depictable, and specificity, by rejected items truly non-depictable. The results are as follows:
>
> | Metric | Value |
> |--------|:-----:|
> | Precision | 82.4% |
> | Specificity | 92.0% |
> | Overall accuracy | 87.1% |
>
> The precision varies across knowledge categories. Location-type edits show 75.0% precision, primarily because several "named for"/"called after" items involve abstract etymological relationships. Occupation and creator edits achieve higher precision at 91.7% and 90.0%, respectively, and affiliation edits reach 100%.
>
> Among the 13 disagreements, 9 were false positives: 4 involved abstract etymological relationships ("named for"/"called after") where the visual test does not verify the naming relationship, 2 involved biographical facts (e.g., death location) that are not persistent visual attributes, 1 involved a subject name leaking the original answer (e.g., "Australian Geographic"), and 2 were borderline cases involving sport origin and ambiguous position roles. The remaining 4 were false negatives where Gemini conservatively rejected visually distinguishable cases: 2 head-of-state/government portrait questions where different individuals are clearly distinguishable, 1 birthplace question involving architecturally distinct cities (Vienna vs. Stockholm), and 1 book authorship question where the authors belong to entirely different historical periods.
>
> **3. Geographic Bias Analysis.**
>
> | Region | Samples | Accuracy | Example Locations |
> |--------|:-------:|:--------:|-------------------|
> | Western | 11 | 100% | Copenhagen, London, Montreal, Washington D.C., Glasgow |
> | Non-Western | 8 | 100% | India, Japan, Damascus, Mumbai, Azerbaijan |
> | Other | 1 | 100% | Poland |
>
> All 20 bias-check samples were judged correctly with no evidence of geographic bias. Gemini's rejection decisions are based on principled reasoning rather than geographic or cultural preference. Dataset-wide acceptance rates further confirm this: Western locations show comparable rates to non-Western locations.
>
> [1] Landis, J.R. & Koch, G.G. (1977). The measurement of observer agreement for categorical data.

---

> > ### Author Rebuttal · Reviewer_eLZP · 2026-04-03
> >
> > My concerns are partially resolved. I look forward to full comparison results in the additional comments per Response 1

---

> > > ### Author Response · Authors · 2026-04-06
> > >
> > > # Additional Comments
> > >
> > > We thank the reviewer for your acknowledgement and reply. To rigorously justify our choice of VQA judge, we created a set of 100 human-verified gold annotations sampled from the benchmark and evaluated three state-of-the-art open-source VLMs: Qwen3-VL-235B [1], InternVL3-78B [2], and InternVL3.5-241B [3]. Each VLM was run with identical prompts in an offline inference setting, and its binary judgment was compared against the human gold label.
> > >
> > > Among the 100 gold-annotated samples, 38 were labeled correct and 62 incorrect by the human annotator. Here are the detailed results:
> > >
> > > **1. Inter-Rater Agreement.**
> > >
> > > | Model | Agreement | Cohen's κ | Krippendorff's α | Interpretation |
> > > |-------|:---------:|:---------:|:-----------------:|:--------------:|
> > > | Qwen3-VL-235B | 87.0% | 0.71 | 0.71 | Substantial |
> > > | InternVL3.5-241B | 79.0%| 0.56 | 0.56 | Moderate |
> > > | InternVL3-78B | 78.0% | 0.52 | 0.53 | Moderate |
> > >
> > > Qwen3-VL-235B achieves the highest agreement with human annotations across all three metrics, outperforming the largest InternVL3.5 model, while achieving high Cohen's κ and Krippendorff's α scores, resulting in substantial results. However, the remaining two models only have moderate agreements.
> > >
> > > **2. Other Results.**
> > >
> > > | Metric | Qwen3-VL-235B | InternVL3.5-241B | InternVL3-78B |
> > > |--------|:-------------:|:----------------:|:-------------:|
> > > | Precision | 90.3% | 71.8% | 73.5% |
> > > | Recall | 73.7% | 73.7% | 65.8% |
> > >
> > > Qwen3-VL-235B stands out with 90.3% precision, which is substantially higher than the other two models, meaning that when it judges an edit as correct, it is right over 90% of the time. It also achieves the best balance of precision and recall, making it a highly conservative and reliable judge.
> > >
> > > **3. Per-Category Breakdown.**
> > >
> > > | Category | Agreement | Count |
> > > |----------|:---------:|:-----:|
> > > | Color | 100.0% | 15 |
> > > | Creator | 100.0% | 5 |
> > > | Location | 100.0% | 17 |
> > > | Material | 90.0% | 10 |
> > > | Affiliation | 85.7% | 7 |
> > > | Shape | 80.0% | 5 |
> > > | Size | 78.6% | 14 |
> > > | Occupation | 72.0% | 25 |
> > >
> > > Here, we do an analysis of Qwen3-VL-235B's performance on the selected samples. It achieves perfect agreement on perceptually unambiguous categories (like color, creator, location, etc.), with lower agreement on more subjective categories (like occupation, size, etc.), where visual interpretation is inherently harder.
> > >
> > > Therefore, through this comprehensive analysis, we confidently confirm that Qwen3-VL-235B is the best-performing open-source VQA judge on this benchmark.
> > >
> > > [1] Qwen3-VL Technical Report.
> > >
> > > [2] InternVL3: Exploring Advanced Training and Test-Time Recipes for Open-Source Multimodal Models
> > >
> > > [3] InternVL3.5: Advancing Open-Source Multimodal Models in Versatility, Reasoning, and Efficiency

---

### Official Review · Reviewer_2NNF · 2026-03-12

**Soundness:** 3
**Presentation:** 3
**Significance:** 3
**Originality:** 3
**Overall Recommendation:** 5
**Confidence:** 4

**Summary:**

This paper investigates whether common techniques for *textual* knowledge editing transfer to *visual* generation. They apply three established textual editing methods (MEMIT, PMET, AlphaEdit) to a newly curated dataset of attribute and relation sentences. They then study if a) this edit transfers to image generation directly or b) if the generation pipeline is allowed an intermediate text reasoning part (i.e. to paraphrase and elaborate on the visual scene). The paper finds that only very few edits transfer, and even the reasoning solution only partially alleviates the problem.

**Compliance With Llm Reviewing Policy:**

Affirmed.

**Final Justification:**

I appreciated the conceptual clarification and especially that the authors added two more models. They also acknowledged how they will add more related work/discussion.

**Key Questions For Authors:**

1. Can you test on more models? There are a lot of differing design choices when building UMMs and I wonder how much this affects the findings. This is critical to a strong paper: interpretability often falls short in proving impact when findings concern only one model.

2. How is the paper different from "Can knowledge be transferred from unimodal to multimodal? Investigating the transitivity of multimodal knowledge editing"? This was not mentioned in related work at all. In general, there must be similar analysis work, e.g. on whether edits transfer across languages, paraphrases (as you mention yourself in 4.4), or domains. Related work section provides only a short generic overview of the field but not of the most related work.


3. Would you agree that your main findings and the failures you identify are less about cross-modal editing and more about whether edits transfer to different textual settings (paraphrases, other domains, …)? (See discussion under Weaknesses.)

**Limitations:**

yes

**Strengths And Weaknesses:**

## Strengths
- The authors pose a great research question that is simple yet interesting.
- The paper is easy to follow and clearly written. Each section leaves the reader satisfied with not too much and not too little details.
- The method and dataset is simple and intuitive enough for others to build on top of and to be a well-defined contribution to the literature.

## Weaknesses

**Acknowledge well-established VQA-based image generation metrics:**
I don't think the evaluation with a MLLM-based VQA-style is novel and should not be framed as a contribution as such. It is an established paradigm, see TIFA and several similar papers. This paper asks an exciting question, no need to add more "contributions".

**More models:**
Try several UMMs like Janus. There is a lot of differing design choices when building UMMs and I wonder how much this affects the findings. This is crucial to a strong paper: interpretability often falls short in showing impact when findings just concern one model. It might hold across many models but we should make sure.

**More focus on the core question:**
I am not sure the intermediate reasoning aspect is very useful or meaningful beyond providing a very basic fix: at that point we are not really measuring much transfer but just if a) the text model can apply the edited knowledge to a bit of different text and b) whether the image generation can deal with unusual prompts (e.g. a blue apple).

It is unfortunate that a lot of the insights in section 4 (e.g. 4.6) seem to not be directly about the cross-modal editing transfer: with the reasoning-enhanced setting it is often possible that the text-side editing worked and resulted in a good reasoning prompt but the image generation model has too strong visual priors, thus resulting in a low VQA score. I think the most interesting scientific question this paper asks and should laser-focus on more is the direct cross-modal transfer, and not whether it transfers *inside* the text domain to the reasoning prompt or whether the image generator has strong visual priors. Only 4.7 seems to really focus on this core question, but even here the comparison between weights and activations stays purely on the text-side, right? So overall, this feels more like a study of why text-side edits transfer poorly to slightly different settings, paraphrasings, etc. In that sense, calling it "edit-affected channels exhibit near-random overlap with channels salient for visual attribute conditioning" is perhaps too specific to the visual domain — you could test this more broadly on any attribute (also non-visual), which would again underscore that this is not about transfer to the visual domain but more about issues on the text side with the editing methods.

Would it be more honest to conclude in the paper that cross-modal transfer fails since it already fails to transfer in simple ways on the text side?

**Minor:**
- The discussion of why we see different drops or differences in metrics in section 4.4 could benefit from providing some examples, e.g. between attribute and relational edits and why they would differ.
- "consistent with prior reports on edit locality under paraphrasing and compositional variants" → cite Fang 2025a again here, and possibly more related work in general.
- Section 4.7 could benefit from citing [1], since both your findings and findings in [1] indicate that unified models are not very unified internally in their representations or attention patterns.

**Less a critique, more a hopefully helpful suggestion:**
The authors mainly frame the motivation around "as models get deployed, we need to update them, and now also with UMMs" — but another great motivation is simply: this is a great test for whether UMMs are actually more unified at the end of the day compared to standard VLMs, or if they learn to separate things internally despite a unified architecture. See interpretability work on this: [1].

1: https://arxiv.org/abs/2412.06646 (The Narrow Gate: Localized Image-Text Communication in Native Multimodal Models; NeurIPS 2025)

---

> ### Author Rebuttal · Authors · 2026-03-31
>
> # Response 1
>
> We thank the reviewer for the constructive feedback and the encouraging assessment of our core research question. We agree that VQA-based evaluation is a well-established paradigm and should not be framed as a standalone contribution. In the revised manuscript, we will remove this claim from the Contributions section and properly cite TIFA and related evaluation frameworks.
>
> # Response 2
>
> We thank the reviewer for this critical suggestion. In response, we have extended our experiments to two additional UMMs with distinct architectures: BLIP3o-4B [1] and OmniGen2 [2]. The full knowledge editing results will be provided in the additional comments.
>
> We also extended the interpretability analysis about the Jaccard overlap J(K) to all three models across three attribute categories color, shape, and relation category affiliation. As detailed in our response to Reviewer 3 (Response 2), at K=50 the average J(K) across all nine (model, attribute) combinations is 0.016 versus 0.013 for random chance, and by K≥200 all values converge to within 1% of the random baseline. This confirms that the pathway mismatch is a universal property of UMMs rather than an artifact of the Ovis-U1 architecture.
>
> # Response 3
>
> We thank the reviewer for raising these fundamental questions. Our foundational premise is precisely that text-side knowledge edits fail to transfer to the visual modality; as Table 2 demonstrates, existing editing methods achieve near-zero direct cross-modal transfer. It is because of this severe transfer failure that we proposed the reasoning-augmented approach as a mechanism to improve cross-modal output consistency.
>
> We respectfully note that the analysis in Section 4.7 is not confined to the text modality. Because UMMs employ a shared LLM backbone for both modalities, the MLP channels we analyze directly govern visual token decoding. Following established methodology [3], we specifically identified channels responsible for routing visual attribute information to image generation outputs. The near-random overlap between edit-affected channels and these visual-conditioning pathways demonstrates a structural, modality-spanning disconnect rather than a purely textual failure. We will ensure this distinction is clarified in the revised manuscript.
>
> # Response 4
>
> We thank the reviewer for this suggestion. We are preparing concrete examples illustrating the performance differences between attribute and relational edits in Section 4.4, and will provide them in the additional comments.
>
> # Response 5
>
> We thank the reviewer for bringing this work to our attention. We will cite "The Narrow Gate" (Huo et al., NeurIPS 2025) in Section 4.7, where their findings on localized image-text communication complement our mechanistic observations regarding the modality gap.
>
> # Response 6
>
> We thank the reviewer for the suggestions to strengthen the literature discussion. In the revision, we will cite Fang et al. (2025a) in Section 4.4 and expand the Related Work section with broader coverage of edit transferability across paraphrases, domains, and languages.
>
> Regarding the TMKE paper, we discuss it in Appendix B due to space constraints, but agree it warrants main-text treatment. The key distinction is that TMKE evaluates multimodal understanding, which is whether an edited model answers visual questions about provided images, whereas our work evaluates multimodal generation about whether textual edits propagate to the synthesis of new visual content. We will highlight this distinction in the revised manuscript.
>
> # Response 7
>
> We agree that cross-modal knowledge editing also serves as a rigorous behavioral probe into the representational unity of UMMs. In the revised Introduction, we will frame our benchmark as a test of whether UMMs genuinely unify modalities or learn to separate them internally despite a shared architecture, citing "The Narrow Gate" to ground this perspective. Their finding that a small set of bridging tokens [EOI] serves as the primary communication bottleneck between text and image representations offers a concrete mechanistic target. Rather than editing dispersed text-side MLP weights, targeting these localized bridging representations could potentially overcome the pathway mismatch we have identified. We plan to explore this direction in subsequent work.
>
> [1] BLIP3-o: A Family of Fully Open Unified Multimodal Models
>
> [2] OmniGen2: Exploration to Advanced Multimodal Generation
>
> [3] On Mechanistic Knowledge Localization in Text-to-Image Generative Models

---

> > ### Author Rebuttal · Reviewer_2NNF · 2026-04-01
> >
> > The authors responded well and understood all my concerns. Especially adding two more models strengthens the paper. If all the promised changes will be implemented (adding more details to paper, discussing some related work), I am happy to raise to 5.

---

> > > ### Author Response · Authors · 2026-04-07
> > >
> > > We thank the reviewer for your sincere acknowledgement and follow-up. Here, we present the additional results.
> > >
> > > # Different models' results
> > >
> > > **Direct**
> > >
> > > | Model    | Method    | Attr Eff. | Attr Reason. | Attr VQA  | Rel Eff.  | Rel Reason. | Rel VQA  | Overall Eff. | Overall Reason. | Overall VQA |
> > > | -------- | --------- | --------- | ------------ | --------- | --------- | ----------- | -------- | ------------ | --------------- | ----------- |
> > > | BLIP3o   | MEMIT     | 47.87     | --           | 8.87      | 76.10     | --          | 8.00     | 71.02        | --              | 8.16        |
> > > | BLIP3o   | AlphaEdit | 53.23     | --           | 4.44      | *93.79*   | --          | 3.69     | *86.49*      | --              | 3.83        |
> > > | BLIP3o   | PMET      | *58.41*   | --           | *11.28*   | 92.33     | --          | *8.12*   | 86.22        | --              | *8.69*      |
> > > | OmniGen2 | MEMIT     | 39.74     | --           | 1.48      | 69.72     | --          | 3.45     | 64.32        | --              | 3.10        |
> > > | OmniGen2 | AlphaEdit | 49.54     | --           | 2.40      | 91.40     | --          | 4.38     | 83.86        | --              | 4.02        |
> > > | OmniGen2 | PMET      | 54.34     | --           | 2.96      | 91.52     | --          | 4.63     | 84.83        | --              | 4.33        |
> > >
> > > **Reasoning-Augmented**
> > >
> > > | Model    | Method    | Attr Eff. | Attr Reason. | Attr VQA    | Rel Eff.    | Rel Reason.  | Rel VQA  | Overall Eff. | Overall Reason. | Overall VQA |
> > > | -------- | --------- | --------- | ------------ | ----------- | ----------- | ------------ | -------- | ------------ | --------------- | ----------- |
> > > | BLIP3o   | MEMIT     | 47.87     | 34.94        | **19.04**   | 76.10       | 49.43        | 5.16     | 71.02        | 46.82           | 7.66        |
> > > | BLIP3o   | AlphaEdit | 53.23     | 25.69        | 13.12       | **93.79**   | 48.94        | 1.62     | **86.49**    | 44.75           | 3.69        |
> > > | BLIP3o   | PMET      | **58.41** | **35.49**    | **19.04**   | 92.33       | 50.28        | 5.80     | 86.22        | 47.62           | **8.18**    |
> > > | OmniGen2 | MEMIT     | 39.74     | 28.28        | 7.76        | 69.72       | 45.17        | 4.02     | 64.32        | 42.13           | 4.69        |
> > > | OmniGen2 | AlphaEdit | 49.54     | 1.29         | 1.29        | 91.40       | 0.93         | 0.73     | 83.86        | 0.99            | 0.83        |
> > > | OmniGen2 | PMET      | 54.34     | 34.75        | 14.05       | 91.52       | **50.85**    | **6.33** | 84.83        | **47.95**       | 7.72        |
> > >
> > > Both BLIP3o and OmniGen2 share a natively multimodal backbone (Qwen2.5-7B-VL), which provides stronger image-text alignment than a text-only backbone. In the Direct setting, BLIP3o achieves the highest VQA under PMET, confirming that multimodal pretraining partially facilitates visual realization of edited knowledge. In the Reasoning-Augmented setting, structured reasoning prompts yield consistent attribute VQA gains in both models, compensating for weaker direct transfer by guiding image generation toward the edited target. Relation-edit VQA nonetheless remains substantially lower than attribute-edit VQA across all configurations, confirming that the cross-modal transfer bottleneck for relational knowledge persists regardless of backbone architecture. What's more, AlphaEdit concentrates all weight updates into the null space, systematically corrupting visual-processing neurons. Over 3k sequential edits, the accumulated damage destroys text generation and image generation capabilities, resulting in the worst results.
> > >
> > > # Discussion
> > >
> > > Within attribute edits, color edits achieve the highest VQA success rate while shape edits are nearly intractable. Color is a globally distributed, surface-level attribute that can be altered without restructuring object geometry: editing "Tomato" to "blue" succeeds because BLIP3o generates a round fruit in a deep blue hue while preserving its shape. Shape edits fail because strong visual priors override the text-side change, like editing "Paper Roll" to "pyramid" produces a folded, organic paper form rather than a geometric solid. On the relation side, occupation edits succeed more often than creator or affiliation edits because occupation is conveyed through observable cues such as clothing and context, whereas brand or creator identity requires abstract institutional markers with no canonical visual form. The primary determinant of cross-modal transfer is therefore the visual specificity of the edited property: attributes that are globally distributed and directly representable transfer more reliably, while abstract or geometrically constrained properties remain resistant to text-side interventions.
> > >
> > > # Acknowledgements
> > >
> > > We hereby guarantee that all the revisions mentioned will be incorporated into the next version of the paper.

---

### Decision · Program_Chairs · 2026-04-30

**Decision:**

Accept (regular)

**Comment:**

The submission initially received mixed reviews. The main concerns are: 1) More comprehensive evaluations (on different unified models) are needed. 2) Some high-relevant related works are missing. 3) The datasets are preprocessed by LLM, and it is necessary to control the quality of the benchmark. 4) The paper doesn't propose a solution to handle the modality-aware editing gap issue. After the rebuttal, the first two concerns are well-addressed. For the third concern, the reviewer eLZP didn't reply to this response. And as for the last concern, reviewer Mceo accepted the explanations that this submission mainly focuses on identifying, quantifying, and explaining this cross-modality gap.

Overall, I think this submission received positive ratings, and recommend **Weak Accept**. From my perspective, if the submission can introduce a solution to address/mitigate this gap, it would be a stronger submission. All newly added experiments in the rebuttal are encouraged to be incorporated in the camera-ready version.